# PlanU: Large Language Model Reasoning through Planning under Uncertainty

**Ziwei Deng**[ab*]**, Mian Deng**[ab*]**, Chenjing Liang**[ab]**, Zeming Gao**[ab]**, Chennan Ma**[ab]**, Chenxing Lin**[ab]**,
Haipeng Zhang**[ab]**, Songzhu Mei**[c]**, Cheng Wang**[ab]**, Siqi Shen**[ab†]

[a]Fujian Key Laboratory of Sensing and Computing for Smart Cities,
School of Informatics, Xiamen University (XMU), China
[b]Key Laboratory of Multimedia Trusted Perception and Efficient Computing, XMU, China
[c]School of Computer, National University of Defense Technology, China
{dengziwei,miandeng,liangcj,zeminggao,chennanma,lincx1123,zhanghaipeng}@stu.xmu.edu.cn,
{siqishen,cwang}@xmu.edu.cn, {sz.mei}@nudt.edu.cn

## Abstract

Large Language Models (LLMs) are increasingly being explored across a range of reasoning tasks. However, LLMs sometimes struggle with reasoning tasks under uncertainty that are relatively easy for humans, such as planning actions in stochastic environments. The adoption of LLMs for reasoning is impeded by uncertainty challenges, such as LLM uncertainty and environmental uncertainty. LLM uncertainty arises from the stochastic sampling process inherent to LLMs. Most LLM-based Decision-Making (LDM) approaches address LLM uncertainty through multiple reasoning chains or search trees. However, these approaches overlook environmental uncertainty, which leads to poor performance in environments with stochastic state transitions. Some recent LDM approaches deal with uncertainty by forecasting the probability of unknown variables. However, they are not designed for multi-step reasoning tasks that require interaction with the environment. To address uncertainty in LLM decision-making, we introduce PlanU, an LLM-based planning method that captures uncertainty within Monte Carlo Tree Search (MCTS). PlanU models the return of each node in the MCTS as a quantile distribution, which uses a set of quantiles to represent the return distribution. To balance exploration and exploitation during tree search, PlanU introduces an Upper Confidence Bounds with Curiosity (UCC) score which estimates the uncertainty of MCTS nodes. Through extensive experiments, we demonstrate the effectiveness of PlanU in LLM-based reasoning tasks under uncertainty.

## 1 Introduction

Large language models (LLMs) have demonstrated remarkable capabilities in various domains, such as reasoning and coding [1, 2, 3]. The success of LLMs in various domains has motivated researchers to apply them to decision-making tasks [4, 5, 6], where an agent selects actions based on the current state in order to achieve specific goals.

Prior work has explored leveraging LLMs for decision-making in three ways: (1) LLMs are used as policies [4, 7], where the LLM is provided with necessary prompts to generate actions. (2) LLMs are used as world models [5], where the LLM is repurposed with appropriate prompts to generate the next state and reward based on a given action. (3) LLMs are used as both the policy and the world model [8, 6, 9], our work belongs to such category as it mimics human decision-making process,

---

[*]Equal contribution

[†]Corresponding author

39th Conference on Neural Information Processing Systems (NeurIPS 2025).

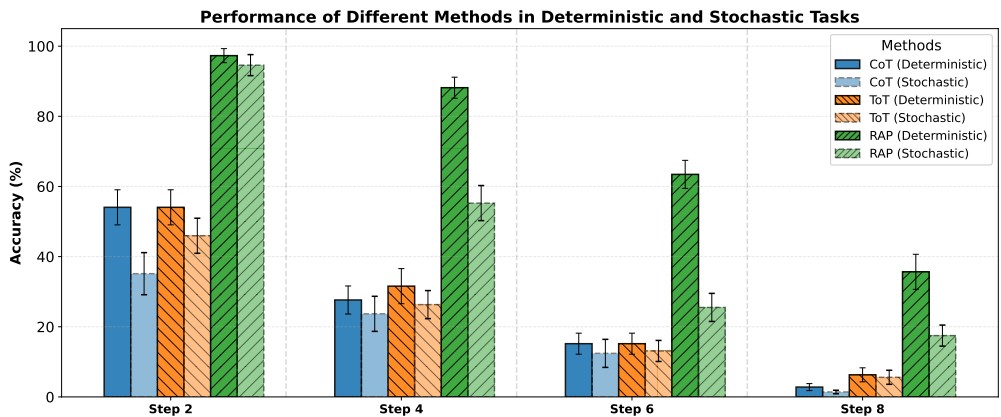

Figure 1: Impact of Environmental Uncertainty on LDM tasks for Block Stacking. The results are grouped according to the least number of steps to finish tasks. *"Deterministic"* indicates deterministic state transitions. In *"Stochastic"* environment, with 20% probability, an action will fail.

simulating and anticipating future outcomes through internal world model and making plans based on it [10].

The adoption of LLMs for decision-making is impeded by uncertainty challenges, such as LLM uncertainty [11, 12] and environmental uncertainty [9]. LLM uncertainty is caused by the stochastic nature of its generation process, which samples the next token through a distribution. A well-known form of LLM uncertainty is hallucinations [13], where LLM generates inaccurate outputs. Most LLM-based decision-making (LDM) approaches address LLM uncertainty by trading off computation for certainty [2, 6, 8]. For example, Self-Consistency [2] samples multiple reasoning traces and selects the most consistent answer. ToT [3] and RAP [8] make plans during search inside trees representing LLM decision spaces.

Most existing LDM approaches overlook environmental uncertainty caused by stochastic transitions. In stochastic environments, the transition between states becomes stochastic rather than deterministic. That is, after taking action $a_t$ in state $s_t$, the environment could transition to multiple possible next states $s_{t+1}$. Moreover, the reward $r_t$ can also be stochastic. As shown in Figure 1, the performance of multiple LLM reasoning approaches drops significantly in the presence of environmental uncertainty. Environmental stochasticity is not rare, even in deterministic environments. State aliasing [14] (e.g., caused by partial observability) could effectively introduce environmental stochasticity [15].

Only a few LDM approaches [9] consider environmental uncertainty. DeLLMa [9] deals with environmental uncertainty by enumerating hidden factors and forecasting their probabilities. However, it does not consider environmental feedback, and thus, it is not suitable for tasks requiring multiple steps of decision-making. In general, integrating LLM-based decision-making methods with uncertainty handling remains challenging. We show that naive integrations, such as prompting uncertainty consideration within LDM, lead to poor performance even on a simple task in Sec 5.2.

In LDM, Monte Carlo Tree Search (MCTS) is widely adopted due to its effectiveness. Our work builds on recent advancements of LLM-based MCTS. When a policy is sampled probabilistically inside an LDM environment, the actions, state transitions, and rewards can cause randomness/uncertainty in the return of the MCTS tree. LLM-MCTS algorithms [8, 6] average the randomness to estimate the return of an MCTS node. *The key insight of our work is to model the return of a node in MCTS as a quantile distribution rather than the mean value.* PlanU builds an MCTS tree consisting of multiple state nodes and action nodes $(s_t, a_t)$, whose return $Z(s_t, a_t)$ is modeled as a quantile distribution [16]. The quantile distribution [16], representing return distribution through quantile, is designed to capture the uncertainty in the returns intrinsic to LDM. A skewed quantile distribution indicates high uncertainty, whereas a uniform quantile distribution indicates the opposite.

During planning, PlanU encourages exploration in the face of uncertainty through an Upper Confidence Bounds with Curiosity (UCC) score. UCC measures curiosity based on the return of actions and uncertainty. Inspired by [17], UCC estimates the environmental uncertainty of a state $s_t$ by comparing features predicted by a neural network to features predicted by a fixed randomly initialized neural network. To address LLM uncertainty that different text descriptions can be generated for the

same state $s_t$, UCC utilizes a text encoder to obtain text embeddings, and then it treats states as the same if their text embeddings are similar.

Through extensive experiments, we demonstrate the effectiveness of PlanU. It performs better than state-of-the-art methods in multiple environments thanks to its ability to deal with uncertainty in LLM decision-making.

## 2 Background

### 2.1 LLM-based Decision Making (LDM) and Uncertainty

In this work, an LDM process is modeled as a Markov Decision Process (MDP). Given a current state $s_t$, an LLM agent generates an action based on a policy $a_t \sim \pi(a|s_t, c_p)$, where $c_p$ is a policy prompt (e.g., task description, in-context example). Once an action is chosen, the state transits from $s_t$ to $s_{t+1}$ according to the state transition dynamics $p(s_{t+1}|s_t, a, c_w)$, where $c_w$ is the world prompt specifying the world dynamics. A reward $r = R(s_t, a_t, c_w)$ is given to the agent.

We focus on two major types of LDM uncertainty: LLM uncertainty [18] and environmental uncertainty [15]. Environment uncertainty is caused by the inherent randomness of the environment. For example, a coin-flipping action could lead to two outcomes (head or tail) of a coin. LLM uncertainty arises from the stochastic nature of its generation process [11], which may produce different responses to the same prompt. For LLM-based policies, such generation uncertainty may lead to incorrect actions or decisions. When using LLM as a world model, LLM uncertainty could lead to incorrect environmental transitions $p(s_{t+1}|s_t, a, c_w)$.

### 2.2 Quantile Distribution

To deal with environmental uncertainty, Distributional Reinforcement Learning [15, 16, 19, 20] models the value of a state-action pair $(s, a)$ as a value distribution $Z(s, a)$. These methods model full value distribution $Z(s, a)$ instead of an expectation value $Q(s, a)$. The value distribution can be modeled (approximated) as a quantile distribution [16, 19] or a categorical distribution [15].

In this work, the quantile distribution $Z(s, a)$ is modeled by quantile functions $\theta$ of a random variable Z, which is defined as follows.

$$\theta_Z(s, a, \tau) = \inf\{z \in \mathbb{R} : \tau \leq F_Z(z)\}, \quad \forall \tau \in [0, 1] \tag{1}$$

where $F_Z(z)$ is the cumulative distribution function of $Z(s, a)$.

Following QR-DQN [16] and IQN [19], we model the value function $Z(s, a)$ as the mixture of $n$ Dirac functions.

$$Z(s, a) = \sum_{i=1}^{n_q} \delta_{\theta(s, a, \tau_i)} p_i(s, a, \tau_i), \tag{2}$$

where $\delta_{\theta(s, a, \tau_i)}$ is a Dirac Delta function whose value is $\theta(s, a, \tau_i)$. $\tau_i$ is a quantile (e.g., 20%). $p_i(s, a, \tau_i)$ is the corresponding probability of $\theta(s, a, \tau_i)$, $n_q$ is the number of quantiles.

### 2.3 Monte Carlo Tree Search (MCTS)

Monte Carlo Tree Search (MCTS) [21, 22] is a widely used decision-making algorithm that explores the search tree smartly. In this tree, nodes represent states, and the edges represent actions. Specifically, an edge $a$ between two nodes $s_1$ and $s_2$, indicating after applying action $a$ on state $s_1$, the state transits to state $s_2$. The standard MCTS assumes a deterministic state transition $p(s_{i+1}|s_i, a) = 100\%$. That's once an action $a$ is taken on $s_i$. It will deterministically transit to a state $s_{i+1}$. Each iteration of a typical MCTS consists of the following four phases:

1. **Selection**: Starting from the root node, at each level $i$ of the MCTS tree, a child node $s_{i+1}$ of the current node $s_i$ is selected. This step ends when a leaf node is reached. The child node $s'$ is selected according to the Upper Confidence applies to Trees (UCT) [23] method.

2. **Expansion**: If the selected node $s_i$ is not a terminal state (leaf node), it is expanded by generating a child state node $s_{i+1}$, which is then added to the search tree.

3. **Simulation**: From the newly expanded state node, multiple simulations or roll-outs are performed until a computation budget is reached and a reward $r$ is obtained.

4. **Back-propagation**: The result of the simulation operation is propagated back to the root node $s_0$.

# 3 Related Work

## 3.1 LLM Reasoning and Decision Making

LLM reasoning decomposes a complex task into multiple intermediate steps to obtain the correct answer/solution. Chain-of-Thought(CoT) [1] and Zero-shot CoT [24] prompt LLMs to generate plans in a step-by-step manner. Least-to-Most [25] decomposes complex problems into a series of sequential sub-questions. Self-Correction [26] performs reasoning through a reasoner and a corrector.

Self-Consistency [2], Repeated Sampling [27] demonstrate that sampling diverse reasoning paths can significantly enhance reasoning effectiveness in tasks such as math reasoning. Researchers [28] find that scaling test-time compute can obtain better performance than scaling model parameters.

Due to the strong reasoning ability of LLMs, they are adopted for decision-making in multiple domains, such as robotics [29, 30], games [31, 7], and web shopping [6]. ReAct [32] extends language models with interaction through external API calls (e.g., web search). Reflexion [33] and Self-Refine [34] explore self-evaluation mechanisms to refine reasoning/decision-making processes.

## 3.2 Tree-based LLM Reasoning and Decision Making

Studies have shown that tree search frameworks significantly improve decision-making performance by breaking down the tasks into multiple steps and searching inside trees wherein each node/edge represents an intermediate step [3, 8, 35].

Tree of Thoughts (ToT) [3] performs breadth-first-search (BFS) or depth-first-search (DFS) to find solutions on trees expanded by LLM. RoT [36] improves tree search performance through reflection. RAP [8] uses a LLM-based policy and a LLM-based world model to perform Monte Carlo Tree Search (MCTS) for decision making. LATS [6] incorporates self-reflection and API-use (e.g., ReAcT) into LLM-based MCTS. TS-LLM[4] is a LLM-based MCTS method with finetuning.

Process Reward Model (PRM) methods have been used in combination of tree search methods, such as STaR [37], RFT [38], ReSTEM [39], V-STaR [40], ReST-MCTS [41] and AlphaLLM [42]. For these methods, reward models assign rewards to each step of the tree search to guide the search.

None of the above tree-based LLM Decision Making/reasoning methods address environmental uncertainty explicitly.

## 3.3 Uncertainty in LLM Decision Making

Multiple methods have been proposed to deal with LLM uncertainty [18], many of them can be categorized into Bayesian inference-based [43, 44], or ensemble-based (e.g., BSDetector [45], and [46]). DeLLMa [9] is a method for LLM decision-making under uncertainty based on classical decision theory. It uses LLM to identify hidden factors and predict their possible outcomes, and then construct utility functions to build a policy that determines the best possible action. However, it does not use environmental feedback (e.g., rewards) to make decisions. Our work can utilize these LLM advancements to improve LLM decision-making.

As far as we know, most existing LLM-based decision-making methods do not explicitly account for environmental uncertainty. The work most relevant to ours is RAP [8], an LLM-MCTS method, which queries the LLM-based world model multiple times to reduce uncertainty and uses the most frequent outcome as the next state during planning. However, it is not specifically designed for environments with uncertainty.

# 4 PlanU: LLM-based Planning Under Uncertainty

PlanU is built upon Monte Carlo Tree Search (MCTS) [47], a powerful planning algorithm that efficiently explores the decision space to obtain high-reward plans. In the vanilla MCTS [22] tree,

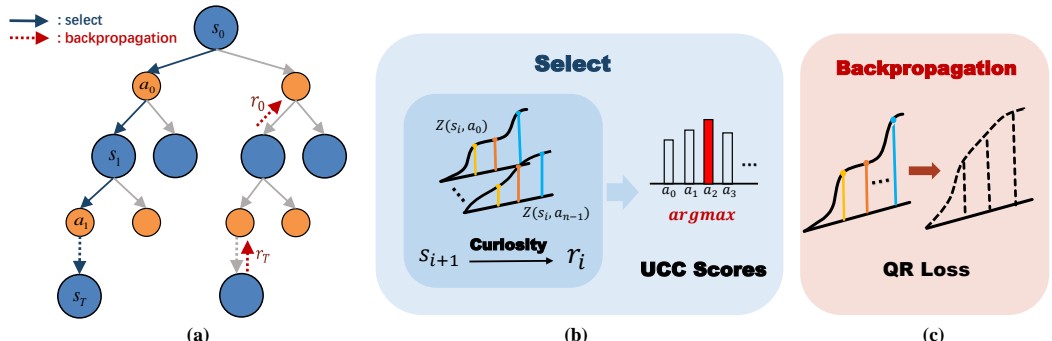

Figure 2: Key stages of PlanU's uncertainty-aware tree search: (a) select and backpropagation phases; (b) UCC-based action selection by combining value distribution $Z(s, a)$ and state novelty, where $\psi[Z(s_t, a_t)]$ maps a quantile distribution to a scalar; (c) distributional updates during backpropagation based on received rewards.

nodes represent states, and edges between nodes represent actions. The value of an edge, denoted as $Q(s, a)$, represents the expected cumulative reward of taking action $a$ in state $s$ and following the optimal policy thereafter.

In stochastic environments, an action may lead to different outcomes. The vanilla MCTS cannot model such scenario. *The core of PlanU is the use of quantile distribution to capture uncertainty within an MCTS tree.* As shown in Figure 2 (a), besides modeling states as nodes, PlanU models action nodes, each of which represents a state-action pair $(s, a)$ in the decision space. PlanU models the return of each action node $Z(s, a)$ as a quantile distribution rather than the expected value $Q(s, a)$ in vanilla MCTS [22].

The quantile distribution $Z(s, a)$ models the overall uncertainty of the state-action pair $(s, a)$. It is used to guide the tree search. An example is given in Figure 2 (b). $Z(s_i, a_0)$ and $Z(s_i, a_{n-1})$ are the quantile distributions of action $a_0$ and $a_{n-1}$ for state $s_i$, respectively. In terms of variance, the return for $a_0$ is more uncertain than that for $a_{n-1}$.

## 4.1 Tree Search under Uncertainty

During the tree search, actions available for a state are added as child nodes of the state node in a search tree. An action could lead to multiple states, which are added as the child node of the action node. Each PlanU iteration consists of four phases, which are described as follows.

**Selection.** Starting from the root node (e.g., the initial state $s_0$), the algorithm selects a child action node based on its UCC values as follows.

$$a^* = \arg \max_{a_t \in c_s(s_t)} UCC(s_t, a_t), \tag{3}$$

where $s_t$ is the current state, $c_s(s_t)$ is the set of actions available for $s_t$, $a_t$ is the action, $UCC(s_t, a_t)$ is the UCC score which will be described in Section 4.2 .

The agent performs the selected action $a_t$ on the state $s_t$, and then the state transits to obtain $s_{t+1}$ according to environment dynamics. If $s_{t+1}$ is not already represented in the search tree, it is added as a child of the action node $a_t$; otherwise, the algorithm transitions directly to the existing node corresponding to $s_{t+1}$. This phase ends upon reaching a leaf state node.

**Expansion.** In this phase, a leaf state node $s_t$ is expanded along with its corresponding action nodes. For each available action $a_t$ in $s_t$, an action node is created. In this work, an action $a_t$ is represented as a sequence of tokens $t_0, t_1 \ldots t_n$. For each action node, we initialize its quantile distribution $Z(s_t, a_t)$ using the probability generated by the LLM, computed as $\pi(s_t, a_t) = \prod_{i=1}^{n} p(t_i|c)$, where $p(t_i)$ denotes the LLM generation probability of token $t_i$, and $c$ is the prompt. Specifically, the quantile distribution is initialized by a set of $n_q$ identical quantile values, each initialized to $\pi(s_t, a_t)$. It is defined as follows.

$$Z(s_t, a_t) = \{\theta_1, \theta_2, \ldots, \theta_{n_q}\} \in \mathbb{R}^{n_q}, \quad \theta_i = \pi(s_t, a_t), \forall i \in \{1, \ldots, n_q\}, \tag{4}$$

where $\theta_i(s_t, a_t, p)$ is the quantile value, and $n_q$ is the number of quantiles. In this way, PlanU can utilize the common sense in LLM to initialize action selection probability.

**Simulation.** This phase simulates multiple trajectories from the newly expanded node to a terminal state to estimate the expected future reward for the current state node $s_t$. Starting from $s_t$, a rollout policy is used to determine the next action and its corresponding next state. In the actual environment, the state resulting from the same action and initial state may vary, making it essential to retrieve the actual next state from the environment at each step. There could be different rollout policies for simulation. For simplicity and efficiency, we use the same policy as the selection phase. A trajectory ends when a terminal state is reached or the depth limit is reached.

**Back-propagation.** Once a terminal state or predefined step limit is reached, a path $\{(s_0, a_0), (s_1, a_1), \ldots, (s_n, a_n)\}$ is traced from the root state node to the terminal state node $s_n$. Along this path, each action node $a_t$ receives a reward $r_t$ provided by the environment. We use Quantile Regression (QR) [16] to update the quantile distribution $z(s_t, a_t)$ of a node. QR estimates the quantiles of the value distribution by minimizing the distance between $Z(s_t, a_t)$ and its target distribution $y(s_{t+1}, a_{t+1}) \triangleq r + \gamma Z(s_{t+1}, a)$. The quantile regression loss is defined as follows.

$$\mathcal{L}_{QR} = \mathbb{E}_{Z_{(s,a)}}[\rho_\tau[y(s_{t+1}, a_{t+1}) - Z(s_t, a_t)]], \qquad (5)$$

where $\tau$ is a quantile sample and $\rho_\omega(\mu) = \mu(\tau - \mathbf{1}\{\mu < 0\})$. Since the quantile regression loss is not smooth, we use Quantile Huber loss [16] to implement $\rho_\tau(\mu)$.

## 4.2 Upper Confidence Bounds with Curiosity

During tree search, PlanU chooses optimism in the face of uncertainty; it encourages exploration through the Upper Confidence Bound with Curiosity (UCC) score. It calculates the uncertainty of text-based states based on the value distribution and novelty of states. Specifically, the UCC score is defined as follows.

$$UCC(s_t, a_t) = \psi[Z(s_t, a_t)] + c_1 \cdot \frac{r_i(s_t)}{N(s_t, a_t)}, \qquad (6)$$

where $\psi[Z(s_t, a_t)]$ is a uncertainty-aware operator that maps a quantile distribution into a real-value. Multiple implementations of $\psi$ can be used. For example, it could be the expectation operator, $\psi[Z(s_t, a_t)] = \mathbb{E}[Z(s_t, a_t)]$ which is the expectation of value distribution of state-action pair $(s, a)$. It could be defined as $\mathbb{E}[Z(s_t, a_t)] + \theta(s_t, a_t, \tau_{0.9}) - \theta(s_t, a_t, \tau_{0.1})$, which consider the spread of return distribution. In default, we use the expectation operator $\mathbb{E}$ as $\psi$. $c_1$ is a hyperparameter, $N(s_t, a_t)$ represents the visit count of the action node $a_t$, and $r_i(s_t)$ is a novelty reward.

The novelty reward $r_i(s_t)$ is designed to encourage the agent to explore less-visited states from a global perspective through estimating the uncertainty of $s_t$. Inspired by [17], $r_i(s_t)$ calculates the feature difference among a predictor network $\hat{f}$ and a target network $f$. Specifically, it is defined as $r_i(s_t) = \left| \hat{f}(e(s_t)) - f(e(s_t)) \right|^2$, where $\hat{f}$ and $f$ are both neural networks.

When estimating the state novelty, to reduce the interference of LLM uncertainty, PlanU uses a text encoder $e(\cdot)$ to map a text-based state $s_t$ to a feature vector. The text encoder can map slightly different text-based descriptions into the similar feature vector, thus reducing text-based uncertainty from LLM. For example, for the same state, LLM could generate a text-based state as "the person is right to the table" or "the table is left to the person". These two texts are semantically the same but different. Through using the text encoder, PlanU can reduce such LLM uncertainty.

The target network $f$ is a neural network that is initialized with random weights, and it is fixed. The parameters of the predictor network $\hat{f}$ are also randomly initialized, but it is different from that of $f$. Plan maintains a buffer $\mathcal{B}$ of the visited states by the agent. After a few iterations, we sample a subset of states from this buffer $\mathcal{B}$, which are then used to train the predictor network $\hat{f}$ through using $r_i(s)$ as the loss, while the target network $f$ remains fixed.

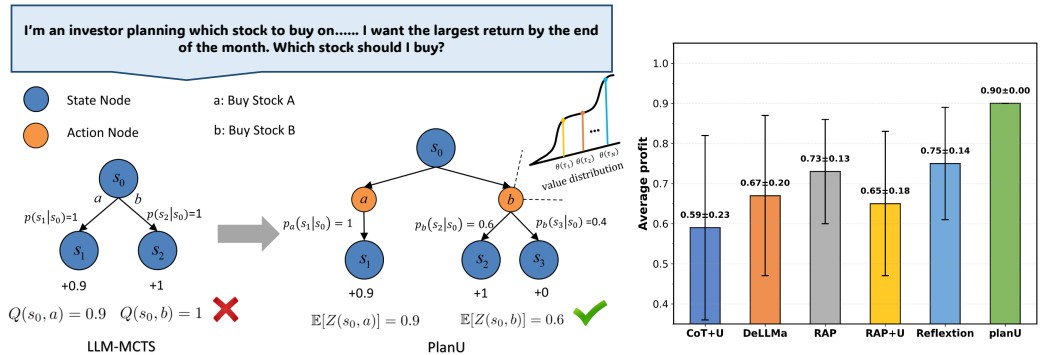

Figure 3: Stock Investment Tasks. The agent faces two actions: buying stock a, which guarantees a profit (reward) of $0.9$, or buying stock b with a 60% probability of earning a profit of $1$ and 40% probability of zero-profit. The right part shows the average profits for different methods.

## 5 Evaluation

We study the performance of PlanU on five decision-making benchmarks: Blocksworld, Overcooked, VirtualHome, Travelplanner and Webshop. We show that PlanU can obtain promising performance across multiple decision-making scenarios under uncertainty. Ablation studies reveal the importance of measuring unceratinty using value distribution and the UCC score for achieving good performance.

### 5.1 Experimental Setup

Our selected baseline methods include:

(1) **CoT** [1]: a prompt-based method that guide LLM to perform chain-of-thought reasoning.
(2) **ToT** [3]: an LLM-based method performs deliberate decision-making inside a search tree.
(3) **Reflexion** [33]: a prompt-based method that considers environmental feedback, which is used to improve its performance.
(4) **DeLLMa** [9]: an LLM-based single-step decision-making method under uncertainty based on classical decision theory.
(5) **RAP** [8]: an LLM-based method that utilizes MCTS for strategic exploration.
(6) **RAP-D**: a variant of RAP. It replaces the MCTS component in RAP by DMCTS [48], which learns a posterior distribution over the utility of the different possible returns.
(7) **RAP-E**: a variant of RAP. It replaces the MCTS component in RAP by EMCTS [49], a method that considers epistemic uncertainty in decision-making.
(8) **LATS**[6]: an LLM-based method that integrates Monte Carlo Tree Search (MCTS) to coordinate reasoning, action, and planning for complex decision-making tasks.
(9) **QR-DQN**[16]: a Reinforcement Learning method using quantile distribution to model uncertainty.

All the experiments are repeated 5 times with different seeds.

### 5.2 A Simple Stock Investment Task

Considering a stock investment task for stock A and B, a person has a fixed profit (i.e., $0.9$) when investing stock A, but has different outcomes (profit of $1$ with 60% chance, otherwise 0) when investing stock B. We use CoT, DeLLMa, RAP, and Reflexion for this task. We have prompted CoT and RAP to consider uncertainty during decision-making, and they are named as CoT+U, RAP+U, respectively. As it is reported in Figure 3, all these methods perform poorly. *This indicates that simple integrations of decision-making method with uncertainty estimation method do not work.*

RAP cannot obtain the best decision. Vanilla MCTS, used in RAP, assumes deterministic environment transitions. To enforce such unrealistic assumption, when determining the child node for a node, MCTS will query the world model multiple times, and use the most frequent state as the child node. Thus, when traveling from $s_0$ through $b$, MCTS will use the most frequent state $s_2$ (with reward 1) as the state, which leads to a suboptimal policy. We show in the left part of Figure 3 that PlanU can learn the correct expected return ($\mathbb{E}[Z(s_0, b)] = 0.6$) while LLM-MCTS (i.e., RAP) fails. Thanks to

Table 1: The Success Rate of the Blocksworld Benchmark. Tasks are organized according to minimal steps ($n$-step) to finish the task.

| Model | Method | Tasks ($n$-step) | | | |
|---|---|---|---|---|---|
| | | **2** | **4** | **6** | **8** |
| Mistral-7B | CoT | 0.514 ± 0.06 | 0.276 ± 0.05 | 0.131 ± 0.03 | 0.000 ± 0.00 |
| | ToT | 0.486 ± 0.06 | 0.355 ± 0.05 | 0.117 ± 0.03 | 0.028 ± 0.01 |
| | RAP | 0.892 ± 0.03 | 0.514 ± 0.06 | 0.166 ± 0.04 | 0.000 ± 0.00 |
| | RAP-D | 0.973 ± 0.02 | 0.632 ± 0.05 | 0.324 ± 0.05 | 0.105 ± 0.03 |
| | RAP-E | **1.000 ± 0.00** | 0.592 ± 0.05 | 0.338 ± 0.05 | 0.084 ± 0.03 |
| | PlanU | **1.000 ± 0.00** | **0.803 ± 0.04** | **0.559 ± 0.04** | **0.217 ± 0.03** |
| LLama3.1-8B | CoT | 0.351 ± 0.05 | 0.237 ± 0.05 | 0.124 ± 0.03 | 0.014 ± 0.01 |
| | ToT | 0.459 ± 0.06 | 0.263 ± 0.05 | 0.131 ± 0.03 | 0.056 ± 0.02 |
| | RAP | 0.946 ± 0.02 | 0.553 ± 0.05 | 0.255 ± 0.05 | 0.175 ± 0.03 |
| | RAP-D | 0.973 ± 0.02 | 0.750 ± 0.04 | 0.428 ± 0.05 | 0.070 ± 0.02 |
| | RAP-E | 0.946 ± 0.02 | 0.763 ± 0.04 | 0.414 ± 0.05 | 0.140 ± 0.03 |
| | PlanU | **1.000 ± 0.00** | **0.842 ± 0.04** | **0.524 ± 0.04** | **0.238 ± 0.03** |
| DeepSeek-R1-Distill-Llama-8B | CoT | 0.405 ± 0.06 | 0.158 ± 0.04 | 0.152 ± 0.04 | 0.077 ± 0.03 |
| | ToT | 0.514 ± 0.06 | 0.237 ± 0.05 | 0.145 ± 0.04 | 0.034 ± 0.02 |
| | RAP | **1.000 ± 0.00** | 0.724 ± 0.04 | 0.200 ± 0.04 | **0.196 ± 0.03** |
| | RAP-D | 0.973 ± 0.02 | 0.750 ± 0.04 | 0.400 ± 0.05 | 0.140 ± 0.03 |
| | RAP-E | **1.000 ± 0.00** | 0.697 ± 0.04 | 0.448 ± 0.05 | 0.175 ± 0.03 |
| | PlanU | **1.000 ± 0.00** | **0.816 ± 0.04** | **0.455 ± 0.05** | **0.196 ± 0.02** |

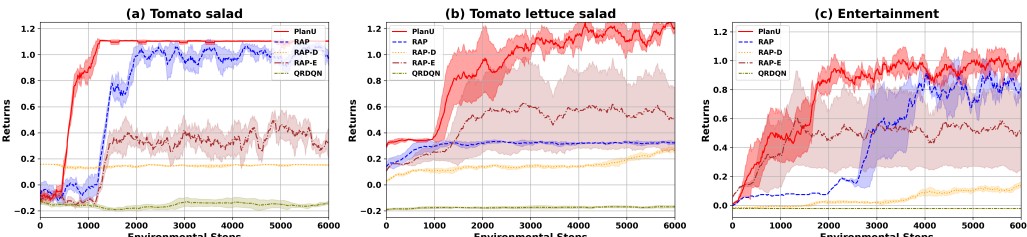

Figure 4: The Return on the Overcooked benchmark: (a) Tomato salad, (b) Tomato lettuce salad, and the VirtualHome benchmark: (c) Entertainment.

the ability to model environmental uncertainty, PlanU can make the optimal decision to obtain the highest profit (i.e., 0.9).

## 5.3 Blocksworld

In Blocksworld tasks [8, 50], the goal is to stack blocks on a table. In this environment, all the states and actions are text-based, and the environmental transition is deterministic. We introduce environmental uncertainty by adding 20% failure rate for actions: STACK, UNSTACK, PUT, and PICKUP. Failed actions keep the state intact. The tasks are categorized into three difficulty tiers according to the minimum required action steps: 2-step (37 tasks), 4-step (76 tasks), 6-step (145 tasks), and 8-step (143 tasks).

We conduct experiments on the Blocksworld tasks on three LLMs: Mistral-7B, Llama3.1-8B, DeepSeek-R1-Distill-Llama-8B. As it is shown in Table 1, PlanU performs the best across all the tasks for the three LLMs. RAP performs better than ToT, which performs better than CoT. The RAP variants, which consider environmental uncertainty, perform the second. PlanU performs better than them, thanks to its ability to handle environmental uncertainty and LLM uncertainty.

## 5.4 Overcooked and Virtual Home

**Overcooked** In the Overcooked environment [51, 7], agents prepare and deliver tomato salad and tomato-lettuce salad using provided resources. The states and actions are text-based. There are two tasks in this environment: Tomato salad, and Tomato lettuce salad. The original environment assumes

Table 2: Experimental results on TravelPlanner

| Method | Task Completion Rate | Constraint Satisfaction Rate |
|---|---|---|
| CoT | 0.156 ± 0.030 | 0.022 ± 0.010 |
| RAP | 0.222 ± 0.025 | 0.044 ± 0.012 |
| LATS [6] | 0.234 ± 0.039 | 0.089 ± 0.017 |
| PlanU | 0.378 ± 0.020 | 0.222 ± 0.015 |

Table 3: Experimental results on WebShop

| Method | Average Reward | Success Rate |
|---|---|---|
| CoT | 0.46 ± 0.04 | 0.1 |
| RAP | 0.41 ± 0.02 | 0.2 |
| LATS [6] | 0.57 ± 0.07 | 0.3 |
| PlanU | 0.73 ± 0.07 | 0.5 |

deterministic transition, we add uncertainty into the environments by adding 20% failure rate for the Chop action. A failed action leads the state unchanged.

The experimental results are presented in Figure 4(a, b), where the horizontal and vertical axes represent environmental steps and returns, respectively. PlanU achieves the best performance across both tasks. RAP and RAP-E rank second on the *Tomato Salad* and *Tomato Lettuce Salad* tasks, respectively. Notably, on the *Tomato Lettuce Salad* task—which is more complex than *Tomato Salad*—PlanU is the only method that successfully completes the task, owing to its strong capability in handling decision-making uncertainty. Compared to RAP-D and RAP-E, PlanU demonstrates greater efficiency in exploring novel states via the UCC score, as well as more effective backpropagation through value distributions, ultimately leading to optimal reasoning paths.

**VirtualHome** The VirtualHome benchmark [52, 7] models a furnished house, where a simulated embodied agent execute actions to complete tasks. We evaluate two tasks: Entertainment (coordinating snack preparation with TV viewing) and Food Preparation (heating a pancake in the microwave). To increase the uncertainty of the tasks, we increase the failure rate for opening appliances (e.g., microwave) as 50%. Failed actions do not impact the environment.

The result for the Entertainment and the Food Preparation task is shown in Figure 4(c) and the Appendix, respectively. These results demonstrate the promising performance of PlanU. PlanU effectively handles task uncertainty, enabling the agent to balance exploitation and exploration, thereby finding the optimal path faster than others. Other methods, including MCTS-based (e.g., RAP) and the RL algorithm (i.e., QRDQN), fail to succeed in this task.

## 5.5 TravelPlanner

We evaluated PlanU on 45 travel planning tasks of varying difficulty using the TravelPlanner benchmark [53]. The benchmark requires agents to generate comprehensive plans covering transportation, meals, and accommodation, with access to search capabilities for information gathering. To simulate real-world uncertainty, delay probabilities were injected for planes and trains based on empirical aviation delay models [54] and national transportation statistics (80.76% on-time rate for flights; 2% delay rate for trains), where delays could invalidate subsequent itinerary segments. Evaluation metrics included Task Completion Rate (coverage of core travel elements) and Constraint Satisfaction Rate (average adherence to commonsense/hard constraints). Results are shown in Table 2.

## 5.6 WebShop

PlanU was evaluated on 10 shopping tasks using the WebShop benchmark [55], where agents must fulfill specific user requirements (e.g., brand, price, size) by interacting with a million-item online store via clicks and searches. To mimic real-world conditions, web actions incurred network latency following a long-tail log-normal distribution(mean 2s), with actions failing if latency exceeded 10s. Performance was measured by average reward (percentage of satisfied user attributes) and success rate (frequency of full requirement fulfillment). Results are presented in Table 3.

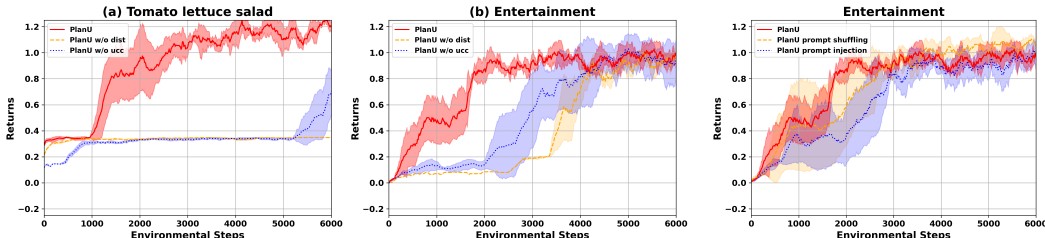

Figure 5: The impact of (a) quantile distribution, (b) UCC score, and (c) LLM uncertainty in PlanU.

## 5.7 Ablation Studies

**Quantile Distribution and UCC Score.** Figure 5(a,b) illustrate the impact of the quantile distribution and the UCC score for PlanU. *PlanU w/o dist* denotes the variant where the quantile distribution (i.e., the first term in Equation 8) is replaced by the mean value, and *PlanU w/o ucc* refers to the variant where the novelty reward (i.e., the second term in Equation 8) is replaced by the UCT exploration term. In *Tomato Lettuce Salad*, removing either the quantile distribution or the UCC term leads to failure in identifying the optimal path. In *Entertainment*, PlanU and its variants can eventually reach the optimal path, PlanU learns faster than its variants. These experiments demonstrate the usefulness of quantile distributions and the UCC scores.

**LLM Uncertainty and Environmental Uncertainty** To evaluate our method's robustness to LLM uncertainty, we introduce two types of perturbations to LLM prompts, which lead to different outputs from LLMs given the same state: (1) **Prompt Shuffling**: We shuffle the sentences in the provided prompt while preserving the sentence boundaries and semantic integrity of the task and environment description. (2) **Prompt Injection**: We inject task-irrelevant but environment-related information into the prompt that does not aid in solving the task. These forms of uncertainty commonly arise when prompts include content generated by LLMs. Examples of such prompts and results for the *Tomato Lettuce Salad* task are provided in Appendix D, while the results for the *Entertainment* task are shown in Figure 5(c). These results demonstrate that LLM uncertainty only slightly affects the convergence speed of PlanU. This indicates that our method is robust to LLM uncertainty. Moreover, we evaluate PlanU against increasing environment stochasticity in the appendix. We find that PlanU performs more robustly than other methods.

## 6 Conclusion

LLM-based decision-making suffers from LLM uncertainty and environmental uncertainty. In this work, we propose PlanU, an LLM-based decision-making method that plans under uncertainty. It is an LLM-based Monte Carlo Tree Search (MCTS) method, where the return value of the MCTS node is modeled as a quantile distribution to capture uncertainty. To balance exploration and exploitation inside the search tree, PlanU uses Upper Confidence Bounds with a Curiosity (UCC) score, which considers the uncertainty of a state. Through extensive experiments, we show that PlanU is a promising method for decision-making under uncertainty.

## 7 Acknowledgments

This work was partially supported by the Fundamental Research Funds for the Central Universities (No. 20720230033), by Xiaomi Young Talents Program. We would like to thank the anonymous reviewers for their valuable suggestions.

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

# Appendix

## A  PlanU Algorithm

### A.1  Algorithm

The PlanU algorithm is described in Algorithm 1.

---

**Algorithm 1** The PlanU Algorithm

---

1: **Require:** Initial state $s_0$, environment dynamics $D$, prior action probability $\pi : S \times A \to \mathbb{R}$, number of iterations $I$, number of quantiles $N_q$, depth limit $L$, prior weight $\alpha$,
2: Initialize memory: $c_a : A \to S$, $c_s : S \to A$, quantile probabilities $\boldsymbol{\tau} = \{\tau_1, \tau_2, \dots, \tau_{N_q}\}$
3: Initialize: value distribution $F : S \times A \to \mathbb{R}^{N_q}$, and state uncertainty networks $\hat{f}, f$
4: EXPANSION($s_0$)
5: **for** $i \leftarrow 1$ to $I$ **do**
6:     $t \leftarrow 0$
7:     **while** $s_t$ is not a leaf state node **do**
8:         $a_t \leftarrow \arg\max_{a \in c_s(s_t)} UCC(s_t, a)$
9:         $(s_{t+1}, \text{done}, r_t) \leftarrow D(s_t, a_t)$
10:        $t \leftarrow t + 1$
11:        **if** $s_{t+1} \notin c_a(a_t, *)$ **then**
12:            $c_a(a_t, s_{t+1}) \leftarrow s_{t+1}$
13:        **end if**                                      ▷ Selection
14:     **end while**
15:     **while** not done and $t < L$ **do**
16:         EXPANSION($s_t$)
17:         $a_t \leftarrow \arg\max_{a \in c_s(s_t)} UCC(s_t, a)$
18:         $(s_{t+1}, \text{done}, r_t) \leftarrow D(s_t, a_t)$
19:         $t \leftarrow t + 1$                                         ▷ Simulation
20:     **end while**
21:     Update uncertainty estimation network $\hat{f}$ with $\{s_1, \dots, s_t\}$
22:     BACK-PROPAGATION
23: **end for**

---

### A.2  Inference Scaling and Vanilla MCTS

Inference-time scaling increases the performance of LLM through increasing computational resources and time used during the inference phase. Multiple inference-time scaling methods are proposed, such as the majority voting, best-of-n, and tree searches. Among them, the Monte Carlo Tree Search (MCTS) technique is one of the most widely utilized tree search methods. It has been used for LLM decision-making and reasoning, such as RAP [8], LATS [6], and TS-LLM [4].

In vanilla MCTS-based LLM planning (i.e., [8]), during planning, the LLM builds a search tree by iteratively selecting the most promising next steps (i.e., actions). It uses a world model to predict future states and rewards. The estimated rewards are used to update the value of the search tree, which improves the planning ability of LLM. Although MCTS-based LLM demonstrated remarkable planning capabilities through the use of MCTS methods, it may fail in simple planning problems with stochastic state transition dynamics.

Vanilla MCTS assumes a deterministic relationship among the next states and actions. When determining the child state node for a state $s_t$ node, the child state node is determined deterministically rather than stochastically once an action $a_t$ is taken from the state $s_t$. Figure 6 illustrates a failure case of an MCTS-based LLM agent for basketball shooting. In MCTS-based LLM [8, 5], when adding a state node into the search tree, an LLM-based world model is queried multiple times, and the most frequently appearing state is used as the state. For Figure 6, the player has a 60% chance of scoring; thus, the scoring state is returned. However, this state transition is not true, which could hurt MCTS performance. The deterministic state transitions assumption is valid in deterministic environments

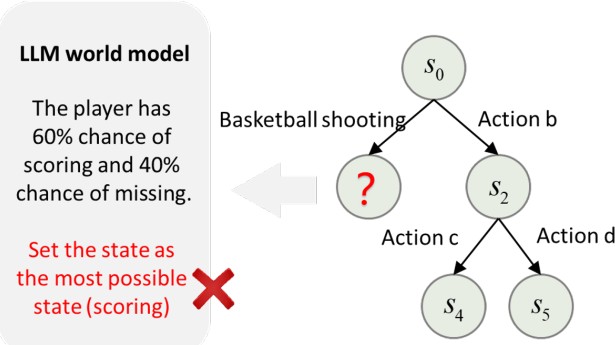

Figure 6: A basketball shooting example, where MCTS fails to model the stochastic state transition. Shooting can lead to the scoring or the missing state, but the vanilla MCTS only models the scoring state.

such as Go but is not valid in real-world environments. Due to the simplified assumption, MCTS performs poorly for stochastic environments.

## B   Baseline Methods and PlanU variants

We consider five categories of methods used for comparison: (1) Prompt-based methods: Chain-of-Though (CoT) and Reflexion; (2) Tree-search methods: ToT, RAP, LATS; (3) Tree-search methods considering uncertainty: RAP-D, RAP-D-UCC, RAP-E; (4) LLM-based decision making method under uncertainty: DeLLMa; (5) Reinforcement Learning method: QRDQN.

1. **CoT** [1]: a prompt-based method that guide LLM to perform chain-of-thought reasoning[1].
2. **ToT** [3]: an LLM-based method performs deliberate decision-making inside a search tree[2]. It searches inside a tree whose nodes represent thoughts.
3. **Reflexion** [33]: a prompt-based method that considers environmental feedback, which is used to improve its performance[3].
4. **DeLLMa** [9]: an LLM-based single-step decision-making method under uncertainty based on classical decision theory[4].
5. **RAP** [8]: an LLM-based method that utilizes MCTS for strategic exploration[5].
6. **RAP-D**: a variant of RAP designed by us. We replaces the MCTS component in RAP by DMCTS [48], which learns a posterior distribution over the utility of the different possible returns.
7. **RAP-D-UCC**: a variant of RAP-D. We combines RAP-D with the UCC scores proposed in this work.
8. **RAP-E**: a variant of RAP designed by us. We replaces the MCTS component in RAP by EMCTS [49], a method that considers epistemic uncertainty in decision-making[6].
9. **LATS** [6]: an LLM-MCTS based decision making method which integrates reflection and API use into the MCTS search process[7].
10. **QRDQN**[16]: a Reinforcement Learning method using quantile distribution to model uncertainty[8].

---

[1]CoT: https://github.com/Ber666/llm-reasoners

[2]ToT: https://github.com/princeton-nlp/tree-of-thought-llm

[3]Reflexion: https://github.com/noahshinn024/reflexion

[4]DeLLMa: https://github.com/DeLLMa/DeLLMa

[5]RAP: https://github.com/Ber666/llm-reasoners

[6]EMCTS: https://github.com/emcts/e-alphazero

[7]LATS: https://github.com/lapisrocks/LanguageAgentTreeSearch

[8]QR-DQN: https://github.com/toshikwa/fqf-iqn-qrdqn.pytorch

In this work, we consider a few PlanU variants to evaluate impact of its design. These variants are listed as follows.

1. **PlanU w/o dist**: a variant of PlanU, where the quantile distribution are removed. It does not capture environmental uncertainty through quantile distribution.

2. **PlanU w/o UCC**: a variant of PlanU, where the UCC scores are removed, the second term in Equation 8 is replaced by the UCT exploration term. It does not fully consider LLM uncertainty.

3. **PlanU prompt shuffling**: a variant of PlanU, where the prompt sentences for the policy and the world model are shuffled, respectively. This is used to evaluate the impact of LLM uncertainty caused by prompt shuffling.

4. **PlanU prompt injection**: a variant of PlanU, its prompts are injected with task-irrelevant but environment-related information. However, these environment-related information are useless for the specific task. This is used to evaluate the impact of LLM uncertainty caused by prompt injection.

## C  Experiment Results

In this experiments, we compare PlanU with 10 different methods, and show that PlanU performs better than all of them in 3 LLM decision making benchmarks. We show that the quantile distribution and the UCC scores play important roles for dealing environmental and LLM uncertainty.

### C.1  Setup and Computing Resources

In this work, all the experiments are repeated with 5 different seeds. For the baseline methods, we use their default configuration. We set the number of quantiles to 50. The $c_1$ in Equation 8 is set to 0.25, and the update rate of the quantile distribution is fixed at 0.5.

The experiments were conducted on high-performance computing clusters equipped with NVIDIA A40 GPU, each with 48GB of memory. The CPUs used in the cluster are Intel(R) Xeon(R) Silver 4216 processors, each running at 2.10GHz. The memory of each computing node is 1024GB.

### C.2  Blocksworld

In Blocksworld tasks [8, 50], the goal is to stack blocks on a table. In this environment, all the states and actions are text-based, and the environmental transition is deterministic. We introduce environmental uncertainty by adding 20% failure rate for actions: STACK, UNSTACK, PUT, and PICKUP. Failed actions keep the state intact. The tasks are categorized into three difficulty tiers according to the minimum required action steps: 2-step (37 tasks), 4-step (76 tasks), 6-step (145 tasks), and 8-step (143 tasks).

#### C.2.1  Task Description

The agent is placed in a Blocksworld environment, where the goal is to stack blocks on a table. Only one block can be moved at a time, and a block cannot be moved if there are other blocks on top of it. Similarly, a block cannot be stacked on another block that already has a block on top. The goal specifies the stacking order, which may include multiple stacks or require some blocks to remain on the table.

**Observation Space:** The environment is a world consisting of a set of blocks, each identified by a color. The agent can only observe the positions of the blocks and their clear status (whether a block is on top of another). The global state includes the positions and statuses of all blocks, and the agent can only observe a limited set of blocks within a defined radius. The initial positions of all blocks are known to the agent. The observation data is directly used as input for planning models such as LLMs, represented through symbolic data.

**Action Space:** The task includes four fundamental actions that the agent can perform:

- **STACK**: The agent stacks a block onto another block.

Table 4: The Success Rate of the Blocksworld Benchmark. Tasks are organized according to minimal steps ($n$-step) to finish the task.

| Model | Method | Tasks ($n$-step) | | | |
|---|---|---|---|---|---|
| | | 2 | 4 | 6 | 8 |
| Mistral-7B | CoT | 0.514 ± 0.06 | 0.276 ± 0.05 | 0.131 ± 0.03 | 0.000 ± 0.00 |
| | ToT | 0.486 ± 0.06 | 0.355 ± 0.05 | 0.117 ± 0.03 | 0.028 ± 0.01 |
| | RAP | 0.892 ± 0.03 | 0.514 ± 0.06 | 0.166 ± 0.04 | 0.000 ± 0.00 |
| | RAP-D | 0.973 ± 0.02 | 0.632 ± 0.05 | 0.324 ± 0.05 | 0.105 ± 0.03 |
| | RAP-E | **1.000 ± 0.00** | 0.592 ± 0.05 | 0.338 ± 0.05 | 0.084 ± 0.03 |
| | PlanU | **1.000 ± 0.00** | **0.803 ± 0.04** | **0.559 ± 0.04** | **0.217 ± 0.03** |
| LLama3.1-8B | CoT | 0.351 ± 0.05 | 0.237 ± 0.05 | 0.124 ± 0.03 | 0.014 ± 0.01 |
| | ToT | 0.459 ± 0.06 | 0.263 ± 0.05 | 0.131 ± 0.03 | 0.056 ± 0.02 |
| | RAP | 0.946 ± 0.02 | 0.553 ± 0.05 | 0.255 ± 0.05 | 0.175 ± 0.03 |
| | RAP-D | 0.973 ± 0.02 | 0.750 ± 0.04 | 0.428 ± 0.05 | 0.070 ± 0.02 |
| | RAP-E | 0.946 ± 0.02 | 0.763 ± 0.04 | 0.414 ± 0.05 | 0.140 ± 0.03 |
| | PlanU | **1.000 ± 0.00** | **0.842 ± 0.04** | **0.524 ± 0.04** | **0.238 ± 0.03** |
| DeepSeek-R1-Distill-Llama-8B | CoT | 0.405 ± 0.06 | 0.158 ± 0.04 | 0.152 ± 0.04 | 0.077 ± 0.03 |
| | ToT | 0.514 ± 0.06 | 0.237 ± 0.05 | 0.145 ± 0.04 | 0.034 ± 0.02 |
| | RAP | **1.000 ± 0.00** | 0.724 ± 0.04 | 0.200 ± 0.04 | **0.196 ± 0.03** |
| | RAP-D | 0.973 ± 0.02 | 0.750 ± 0.04 | 0.400 ± 0.05 | 0.140 ± 0.03 |
| | RAP-E | **1.000 ± 0.00** | 0.697 ± 0.04 | 0.448 ± 0.05 | 0.175 ± 0.03 |
| | PlanU | **1.000 ± 0.00** | **0.816 ± 0.04** | **0.455 ± 0.05** | **0.196 ± 0.02** |

- **UNSTACK**: The agent unstack a block from another block.

- **PUT**: The agent places a block on the table.

- **PICKUP**: The agent picks up a block from the table.

Each action can only be executed if the preconditions are met (e.g., the block is clear or the agent is holding the block). The agent must learn to sequence these actions in order to achieve the goal.

**Dynamics:** The transitions in this environment are deterministic but with some stochasticity introduced in the Stochastic Blocksworld setting. In this setting, the four fundamental actions (STACK, UNSTACK, PUT, PICKUP) have a 20% chance of failure during execution. If an action fails, the state remains unchanged, introducing uncertainty into the state transitions.

**Reward:** A sparse reward setting is used in which the agent only receives a +1 reward upon completing the goal. The task completion is defined as achieving the specified stacking order or leaving certain blocks on the table as required.

**Episode Termination:** Each episode terminates when the agent successfully completes the goal or reaches the maximum number of time steps. If the agent reaches the goal before the maximum time steps, it is rewarded, and the episode ends. If the maximum time steps are reached without completing the task, the episode ends without a reward.

### C.2.2 Experimental Results for Blocksworld

We conduct experiments on the Blocksworld tasks on three large language models. They are Mistral-7B, Llama3.1-8B, and DeepSeek-R1-Distill-Llama-8B. The experimental results are shown in Table 4. Some of the results are already shown in the main paper, for completeness and ease of reading, we have placed the results together.

As it is shown in Table 4, PlanU performs the best across all the tasks for the three LLMs. RAP performs better than ToT, which performs better than CoT. As DeLLMa does not consider environment feedback, thus it is not suitable for this task that requiring multiple steps of decision making. We have compared our approach against the four MCTS-based LLM approaches: RAP, RAP-D, RAP-E. The RAP variants (i.e., RAP-D, RAP-E), which consider environmental uncertainty, perform better than RAP. PlanU performs better than them, thanks to its ability to handle environmental uncertainty and LLM uncertainty.

**Closed-Source and Open-Source Larger Model Evaluation in Blocksworld**

We further evaluated larger models in Blocksworld, including Qwen/Qwen2.5-14B-Instruct (open-source), OpenAI GPT-4.1 (closed-source), and Google Gemini-2.5-Pro (closed-source). These models cover both large-scale open-source and closed-source paradigms, enabling a comprehensive assessment of our method's generalization across model types. The experimental results are summarized in Tables 56 and 7. It can be observed that PlanU achieves superior performance consistently on both large-scale open-source and closed-source models, verifying its strong adaptability to different model architectures.

Table 5: Performance Comparison on OpenAI GPT-4.1 (Closed-Source) in Blocksworld

| Method | 2-step | 4-step | 6-step | 8-step |
|---|---|---|---|---|
| CoT | 0.541 ± 0.07 | 0.382 ± 0.06 | 0.221 ± 0.05 | 0.063 ± 0.04 |
| ToT | 0.595 ± 0.08 | 0.421 ± 0.07 | 0.241 ± 0.06 | 0.077 ± 0.05 |
| RAP | 1.000 ± 0.00 | 0.763 ± 0.05 | 0.341 ± 0.03 | 0.231 ± 0.06 |
| PlanU | 1.000 ± 0.00 | 0.821 ± 0.04 | 0.648 ± 0.03 | 0.392 ± 0.04 |

Table 6: Performance Comparison on Google Gemini-2.5-Pro (Closed-Source) in Blocksworld

| Method | 2-step | 4-step | 6-step | 8-step |
|---|---|---|---|---|
| CoT | 0.568 ± 0.08 | 0.395 ± 0.07 | 0.228 ± 0.06 | 0.056 ± 0.05 |
| ToT | 0.622 ± 0.07 | 0.408 ± 0.08 | 0.248 ± 0.07 | 0.070 ± 0.06 |
| RAP | 1.000 ± 0.00 | 0.776 ± 0.03 | 0.328 ± 0.04 | 0.238 ± 0.05 |
| PlanU | 1.000 ± 0.00 | 0.808 ± 0.05 | 0.662 ± 0.03 | 0.378 ± 0.04 |

Table 7: Performance Comparison on Qwen/Qwen2.5-14B-Instruct (Open-Source) in Blocksworld

| Method | 2-step | 4-step | 6-step | 8-step |
|---|---|---|---|---|
| CoT | 0.405 ± 0.06 | 0.178 ± 0.05 | 0.138 ± 0.04 | 0.028 ± 0.03 |
| ToT | 0.486 ± 0.05 | 0.247 ± 0.06 | 0.131 ± 0.05 | 0.042 ± 0.04 |
| RAP | 0.973 ± 0.02 | 0.605 ± 0.03 | 0.219 ± 0.03 | 0.161 ± 0.05 |
| PlanU | 1.000 ± 0.00 | 0.750 ± 0.02 | 0.521 ± 0.05 | 0.182 ± 0.04 |

**Comparing with LLM-based Decision-making method under uncertainty: DeLLMa**

We evaluate DeLLMa in the Blocksworld environment with stochastic state transitions on Mistral-7B. As DeLLMa is not specifically designed for multi-step decision-making, we treat each planning task as a single-step decision problem. The prompt for DeLLMa is depicted in Appendix D. DeLLMa performs multi-step reasoning by integrating recent inference-time techniques grounded in decision and utility theory to produce accurate and auditable outcomes.

As shown in Table 8, DeLLMa outperforms CoT and ToT in short-horizon tasks (e.g., Step 2), demonstrating its effectiveness in simple reasoning scenarios. However, its performance deteriorates significantly as the number of required steps increases, primarily due to its heavy reliance on the LLM's reasoning ability—where uncertainty compounds over longer reasoning chains—and its lack of explicit modeling for multi-step decision-making (lack of considering environmental feedback).

Our method PlanU maintains consistently high accuracy across all steps, demonstrating superior capability in handling complex, long-horizon decision-making tasks.

**Comparing with LLM-based MCTS Decision-making method: LATS**

We evaluate the performance of LATS in the stochastic Blocksworld environment on Mistral-7B. As shown in Table 9, LATS consistently outperforms RAP across different planning horizons. This improvement can be attributed to its *reflection* mechanism, which allows the language model to reason over its past trajectories and revise future actions accordingly. Such a mechanism indeed enhances robustness in the face of uncertainty. However, compared to our approach, which leverages value distribution modeling for uncertainty estimation, LATS still lacks the precision required

Table 8: The Success Rate of the Blocksworld Benchmark: DeLLMA, CoT, and ToT

| Method | Step 2 | Step 4 | Step 6 | Step 8 |
|--------|--------|--------|--------|--------|
| CoT | 0.514 ± 0.062 | 0.276 ± 0.051 | 0.131 ± 0.027 | 0.000 ± 0.000 |
| ToT | 0.486 ± 0.062 | 0.355 ± 0.055 | 0.117 ± 0.026 | 0.028 ± 0.014 |
| DeLLMa | 0.595 ± 0.081 | 0.092 ± 0.033 | 0.041 ± 0.017 | 0.000 ± 0.000 |
| PlanU | 1.000 ± 0.000 | 0.803 ± 0.045 | 0.559 ± 0.041 | 0.217 ± 0.034 |

Table 9: The Success Rate of the Blocksworld Benchmark: LATS and RAP variants

| Method | Step 2 | Step 4 | Step 6 | Step 8 |
|--------|--------|--------|--------|--------|
| RAP | 0.892 ± 0.050 | 0.513 ± 0.057 | 0.166 ± 0.031 | 0.000 ± 0.000 |
| RAP-D | 0.973 ± 0.022 | 0.632 ± 0.053 | 0.324 ± 0.049 | 0.105 ± 0.030 |
| RAP-E | 1.000 ± 0.000 | 0.592 ± 0.054 | 0.338 ± 0.052 | 0.084 ± 0.031 |
| LATS | 0.946 ± 0.037 | 0.684 ± 0.053 | 0.352 ± 0.040 | 0.077 ± 0.023 |
| PlanU | 1.000 ± 0.000 | 0.803 ± 0.045 | 0.559 ± 0.041 | 0.217 ± 0.034 |

for long-horizon planning. The performance gap becomes more pronounced as the number of steps increases, highlighting the limitations of reflection-based reasoning in capturing fine-grained stochastic dynamics.

## C.3 Overcooked

In the Overcooked environment [51, 7], agents prepare and deliver tomato salad and tomato-lettuce salad using provided resources. The states and actions are text-based. There are two tasks in this environment: Tomato salad, and Tomato lettuce salad. The original environment assumes deterministic transition, we add uncertainty into the environments by adding 20% failure rate for the Chop action. A failed action leads the state unchanged.

### C.3.1 Task Description

The agent is placed in the Overcooked kitchen and aims to cook a specified dish using the provided ingredients and tools, delivering it to the "star" cell as soon as possible. The agent must learn the correct procedure, including picking raw vegetables, chopping them, combining the chopped ingredients in a bowl, and then delivering the dish.

**Observation Space:** The environment is a 7×7 grid world that includes ingredients, bowls, cutting boards, and a delivery counter. In the tomato salad task, the environment provides one tomato, one bowl, one cutting board, and one delivery counter. In the tomato-lettuce salad task, the environment provides one tomato, one lettuce, one onion, one bowl, two cutting boards, and one delivery counter. The onion serves as an interference, though it does not appear in the recipe. The global state information includes the positions of the agent and the objects, as well as the status of each ingredient (chopped or unchopped). The environment is partially observable, and the agent can only observe the positions and statuses of objects within a 5×5 square centered around the agent. Other unseen objects are masked in the observation. The initial positions of all objects are known to the agent. The raw observation data is converted into prompts through scripts, which are then used as input for the LLM.

**Action Space:** This work mainly focuses on using high-level macro-actions, as they usually have richer semantics. Each macro-action may take several time steps to complete. The following is a brief description of the macro-actions used in this study:

- **Chop:** The agent stands next to a cutting board with an unchopped ingredient and chops it into pieces.

- **Get-Tomato, Get-Lettuce, Get-Onion, Get-Bowl, Go-Cutting-Board-1, and Deliver:** These actions navigate the agent to the location of the corresponding object and execute the corresponding operation. In the tomato salad task, the available macro-actions are Get-Tomato, Get-Bowl, Go-Cutting-Board-1, and Deliver. In the tomato-lettuce salad task, all macro-actions are valid.

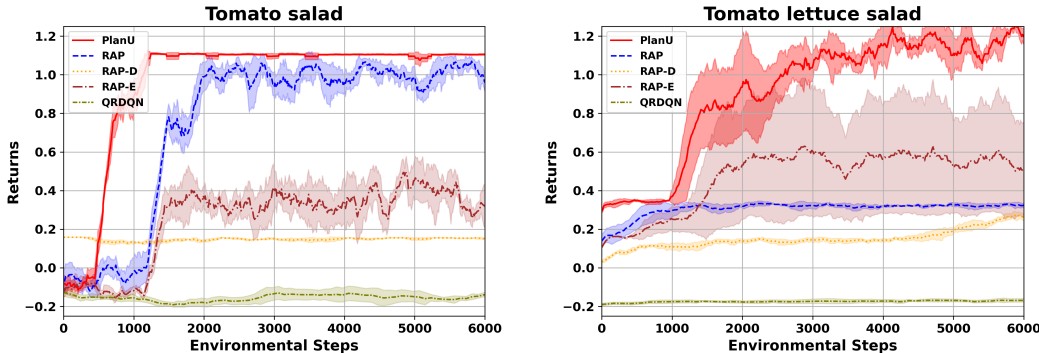

Figure 7: Experimental Results for Overcook: Tomato salad (left) and Tomato lettuce salad (right) in Overcooked.

**Dynamics:** The transitions in this task are deterministic with some randomness. The Chop action has a 20% chance of failing, in which case the state remains unchanged. If the agent delivers the wrong item, that item will be reset to its initial position.

**Reward:** +0.2 for chopping a correct ingredient, +1 terminal reward for delivering the correct dish, -0.1 for delivering the wrong dish, and -0.001 for every time step.

**Episode Termination:** Each episode terminates when the agent successfully delivers the target dish to the delivery counter or reaches the maximum time steps (200).

### C.3.2  Experimental Results for Overcooked

For ease of reading, we have partitioned the experimental results into Figure 7 and Figure 8. The experimental results for PlanU, RAP, RAP-D, RAP-E, and QRDQN are presented in Figure 7, while the results for RAP-D-UCC, CoT, and ToT are shown in Figure 8. In these figures, the horizontal and vertical axes represent environmental steps and returns, respectively. *PlanU achieves the best performance across both tasks.*

In Figure 7, RAP and RAP-E rank second on the *Tomato Salad* and *Tomato Lettuce Salad* tasks, respectively. Notably, on the *Tomato Lettuce Salad* task—which is more complex than *Tomato Salad*—PlanU is the only method that successfully completes the task, owing to its strong capability in handling decision-making uncertainty. Compared to RAP-D and RAP-E, PlanU demonstrates greater efficiency in exploring novel states via the UCC score, as well as more effective backpropagation through value distributions, ultimately leading to optimal reasoning paths.

In Figure 8, the results for CoT, ToT, and RAP-D-UCC. For COT and TOT, we adopt the same base prompt as used in PlanU, augmented with task-specific guiding prompts to enhance LLM reasoning. For each task, we allow the LLM to complete it within 15 steps and evaluate its performance over 10 trials on the same task. The reported result is the average reward across these trials. RAP-D-UCC is a RAP variant designed by us for this task. It considers environmental uncertainty by using DMCTS [48] and LLM uncertainty through using the UCC score (the state novelty part). As it is shown in the left part of Figure 8, it performs inferior to RAP-D. However, it performs better than RAP-D in the right part of Figure 8. This indicates that *a simple integration of methods deal with environmental uncertainty and LLM uncertainty does not work effectively.*

### C.4  VirtualHome

The VirtualHome benchmark [52, 7] models a furnished house, where a simulated embodied agent execute actions to complete tasks. We evaluate two tasks: Food Preparation (heating a pancake in the microwave) and Entertainment (coordinating snack preparation with TV viewing). To increase the uncertainty of the tasks, we increase the failure rate for opening appliances (e.g., microwave) as 50%. Failed actions do not impact the environment.

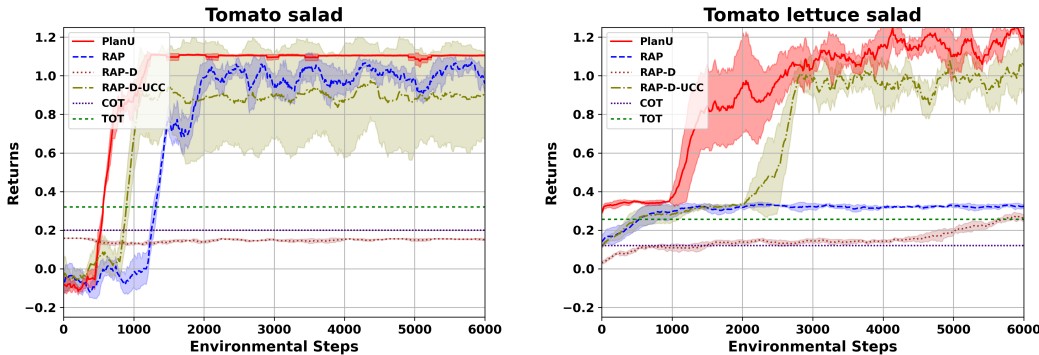

Figure 8: Experimental Results for Overcooked benchmark: Tomato salad (left) and Tomato lettuce salad (right).

### C.4.1 Task Description

The agent, represented as a humanoid avatar, is placed in a fully furnished household with various rich objects to interact with. For the **Food Preparation** task, the agent needs to find the pancake on the table in the kitchen and place it in the microwave to heat, this is a relative simple task. For the **Entertainment** task, the agent needs to find and bring chips and milk from the kitchen to the living room and succeed in sitting on the sofa with the TV on and the chips and milk nearby. The challenge arises when the agent, already holding both the milk and chips, lacks an additional hand to turn on the TV. Consequently, the agent needs to learn to place at least one item on the nearby coffee table before operating the TV.

**Observation Space:** The environment is partially observable. The agent can only see the objects in the current room and cannot see the objects in the other room. The observation consists of a set of Boolean values, representing whether the agent sees the relative object, whether these objects are close to the agent, and the status of the objects, such as whether the TV is on and whether the milk is on the coffee table. The symbolic raw observations are converted to prompts with scripts to serve as input for the LLMs.

**Action Space:** This work mainly focuses on using high-level macro-actions, as they usually have richer semantics. Each macro-action may take several time steps to complete. The following is a brief description of the macro-actions used in this study:

- **Get-Pancake**: The agent picks up the pancake from the table in the kitchen.
- **Place-Pancake-Microwave**: The agent places the pancake into the microwave and closes the microwave door.
- **Get-Chips**: The agent retrieves the chips from the kitchen.
- **Get-Milk**: The agent retrieves the milk from the kitchen.
- **Place-Item-Coffee-Table**: The agent places an item on the coffee table.
- **Sit-On-Sofa**: The agent sits on the sofa.
- **Turn-On-TV**: The agent turns on the TV.
- **Walk-to-A place**: The agent walks to a specific room including the living room, the bedroom, the kitchen and the bathroom.

**Dynamics:** The transitions in this task involve some randomness. In the food preparation task, due to its relative simplicity, there is a 50% failure probability when the agent attempts to open the microwave. In contrast, in the entertainment task, there is a 20% failure probability when the agent tries to get chips or get milk.

**Reward:** We adopt a sparse reward setting in Food preparation, where the agent only receives a +1 reward upon completing the task. In Entertainment, there are three subtasks: get chips, get milk and turn on the TV. The agent will get a +0.1 reward upon completing each of the sub task and a +1 reward if the agent sits on the sofa while every subtask is finished.

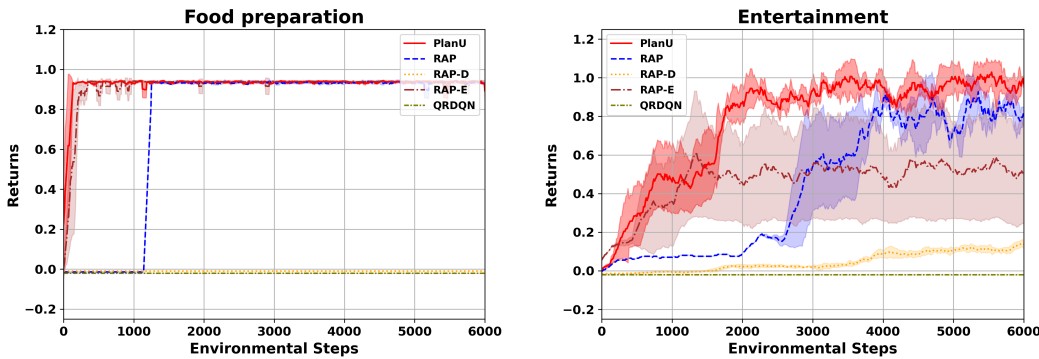

Figure 9: Experimental Results for VirtualHome: Food preparation (left) and Entertainment(right).

**Episode Termination:** Each episode terminates when the agent successfully completes the task or reaches the maximum time steps (50), as each macro-action takes one time step to execute. In the pancake heating task, the agent succeeds when it places the pancake in the microwave and closes the microwave door. In the TV watching task, the agent succeeds when it sits on the sofa, the TV is on, and the chips and milk are on the coffee table or held in hand.

### C.4.2 Experimental Results for VirtualHome

Figure 9 shows that for both the Food preparation and the Entertainment tasks, *PlanU performs the best for these two tasks*. The left part of Figure 9 presents the performance of each algorithm on the *Food Preparation* task. As this task is relatively simple for large language models (LLMs), LLM-based methods are able to quickly discover the optimal path while RL method QRDQN fails to reach the optimal path in limited interactions with the environment. The Entertainment task is more difficult than the Food Preparation task, As it is shown in the right part of Figure 9, RAP performs the second.

### C.5 Token Usage

PlanU is a MCTS-based LLM decision making task under uncertainty. As MCTS is computational intensive, in this section, we evaluate the computational resource consumption of PlanU through using the Token Usage and the Query times metrics. **Token usage** is the number of tokens consumed during the task, while **Query times** refers to the number of time the agent queries a LLM.

We evaluate the performance of PlanU and its variants, RAP, CoT and, ToT over 10 trials on the Tomato Lettuce Salad task. For CoT and ToT, we prompt the LLM to find the optimal path within 15 steps. The Token Usage and the Query time are reported in Table 10.

PlanU consumes the least number of tokens and a number of queries among MCTS methods (e.g., RAP), which demonstrates its ability to explore decision space and obtain the optimal path with less resource consumption. PlanUs consume nearly 3 times less tokens than RAP, a vanilla MCTS-based decision making method.

Although PlanU consumes more tokens than CoT and ToT, it can find correct plan uncertainty, whereas CoT and ToT fail most of the time for the three benchmarks evaluated in this work.

Table 10: Token usage for the Tomato Lettuce Salad task

| Method | PlanU | PlanU w/o dist | PlanU w/o UCC | RAP | COT | TOT |
|---|---|---|---|---|---|---|
| Token Usage(k) | 922.69 ± 6.6 | 1197.99 ± 7.6 | 1521.86 ± 2.9 | 2643.84 ± 8.9 | 35.98 ± 3.2 | 159.85 ± 4.6 |
| Query times | 1389.6 ± 10.0 | 2353.0 ± 14.9 | 2289.2 ± 4.5 | 4313.0 ± 14.8 | 150.0 ± 0.0 | 150.0 ± 0.0 |

## C.6 Ablation Study

We evaluate the impact of the quantile distribution and the UCC score. The results are presented in Figure 10. **PlanU w/o dist** refers to the variant of PlanU where the quantile distribution is replaced with the mean value. **PlanU w/o ucc** denotes the variant without the UCC score, in which the exploration term is replaced by that used in UCT.

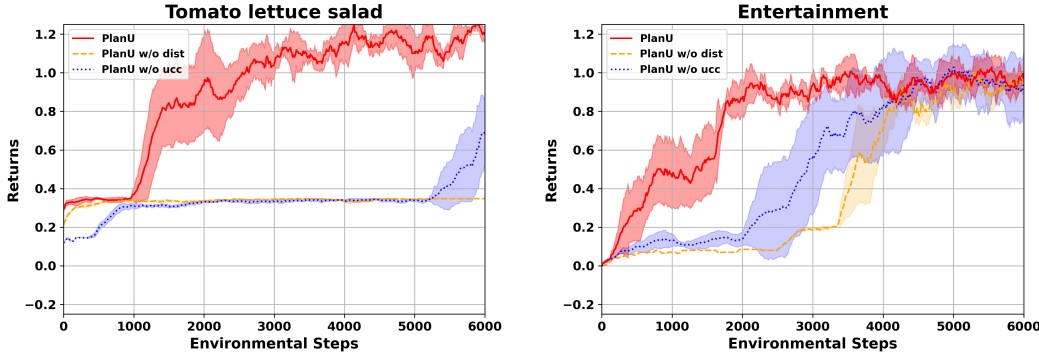

Figure 10: The impact of quantile distribution and UCC score.

### C.6.1 Experimental Results on Operator Ablations

Learning an approximate distribution rather than a single expected value can preserves the multimodality in value distribution, which can lead to more stable learning. As we can see that PlanU using the mean value without the quantile distribution lead to significantly poor performance than PlanU in the following table. Moreover, we have conducted experiments to evaluate the impact of different operators: $\tau_{0.5}$, $\mathbb{E}[Z(s_t, a_t)]$ +variance, and $\mathbb{E}[Z(s_t, a_t)]+ \tau_{0.9} - \tau_{0.1}$. $\mathbb{E}[Z(s_t, a_t)]$ is the mean of the distribution, $\tau_{0.5}$, $\tau_{0.9}$ and $\tau_{0.1}$ are the value for quantile 0.5, 0.9 and 0.1, respectively. As it is shown as follows. In general, using the mean operator can lead to first or second performance.

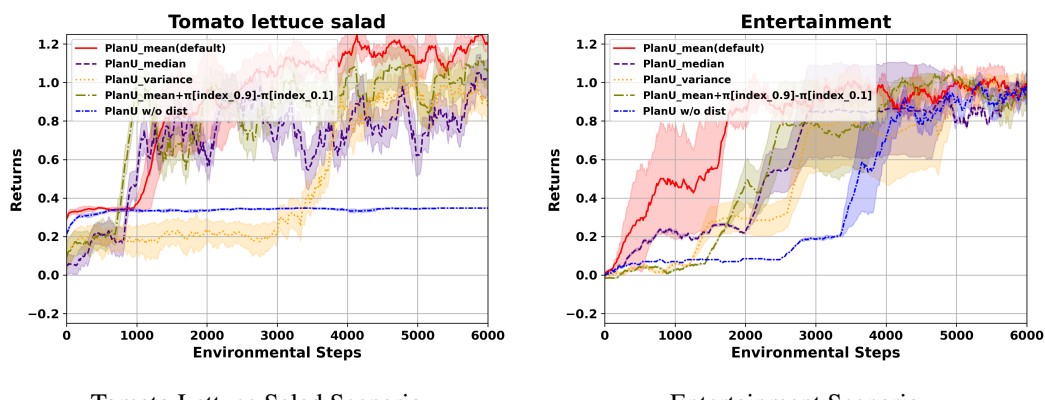

Tomato Lettuce Salad Scenario          Entertainment Scenario

Figure 11: The impact of different operators ($\tau_{0.5}$, $\mathbb{E}[Z(s_t, a_t)]+$ variance, and $\mathbb{E}[Z(s_t, a_t)]+(\tau_{0.9} - \tau_{0.1})$) on performance. Results confirm that preserving multimodality via quantile distributions enhances stability, while the mean operator consistently achieves top-tier performance.

### C.6.2 Ablation Study on Sentence-Transformer Models

To investigate the impact of different sentence embedding models on performance across distinct task settings, we conduct ablation experiments using three widely adopted Sentence-Transformer models. The evaluated models include:

- `sentence-transformers/all-mpnet-base-v2` (used as the default model in our main experiments)

- `sentence-transformers/all-MiniLM-L6-v2`

- `sentence-transformers/all-MiniLM-L12-v2`

We evaluate these models on two representative tasks: the **Tomato Salad** task and the **Tomato Lettuce Salad** task. The results, visualized in Figure 12, demonstrate how embedding model choices affect performance metrics (e.g., success rate, efficiency) across the two tasks.

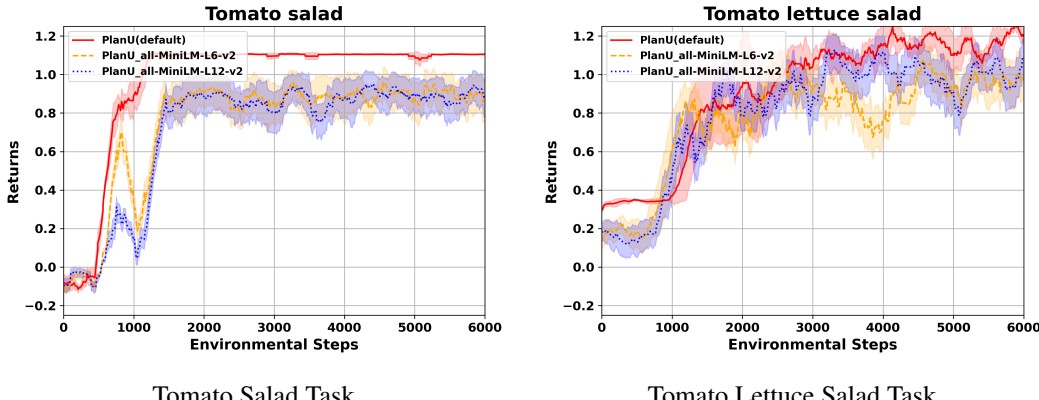

Figure 12: Performance comparison of different Sentence-Transformer models on the Tomato Salad and Tomato Lettuce Salad tasks. The results highlight variations in task adaptation based on model architecture (e.g., parameter size, layer count).

As shown in Figure 12, the default `all-mpnet-base-v2` model generally achieves superior performance, particularly on the more complex Tomato Lettuce Salad task, which may benefit from its larger capacity. Meanwhile, the lighter `all-MiniLM` variants exhibit competitive results in the simpler Tomato Salad task, suggesting a potential trade-off between model efficiency and task complexity.

### C.6.3 Ablation Study on number of quantiles

To examine the sensitivity of our method to the number of quantiles used in modeling the value distribution, we conduct ablation experiments with three different number of quantiles: 41, 51 (default), and 61. These experiments are performed across both the Tomato Salad and Tomato Lettuce Salad tasks, with results visualized in Figure 13.

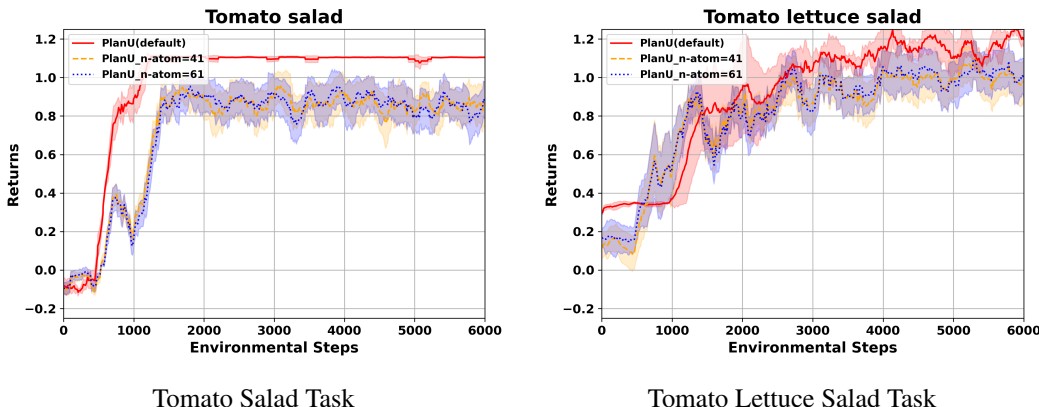

Figure 13: Performance comparison across different number of quantiles (41, 51, 61) on the Tomato Salad and Tomato Lettuce Salad tasks.

As observed in Figure 13, the default setting of 51 quantiles (n=51) outperforms both n=41 and n=61 in most iterations, particularly in later stages. Performance differences across quantile choices are minor, indicating that the method is robust to changes in this parameter, with the default providing

the best results in our experiments. This robustness suggests that the approach does not require fine-grained tuning of the quantile count, simplifying its practical deployment.

## C.7 Impact of Uncertainty

In this section, we evaluate whether PlanU matches the design goal that making good decision under uncertainty. We evaluate the impact of environmental uncertainty for PlanU in Appendix C.7.1, and the impact of LLM uncertainty in Appendix C.7.2. We find that PlanU can deal with environmental uncertainty well thanks to the use of quantile distributions; that PlanU is robust in the face of LLM uncertainty with different prompts and temperatures.

### C.7.1 Impact of Environment Uncertainty

Figure 14 shows the performance for PlanU and PlanU w/o dist, a PlanU variant without value distribution under increased environmental randomness. With increased randomness (increased Action Failure Rate), the success rate of both methods decreases. With quantile distribution, PlanU enjoys a higher success rate than without.

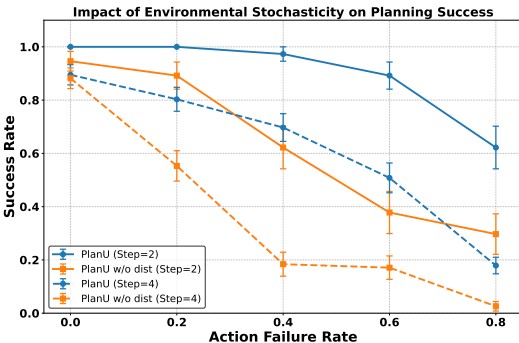

Figure 14: Impact of Environmental Uncertainty for PlanU and PlanU w/o dist. The horizontal axis depicts the action failure rate (AFR). The increase of the AFR lead to the increase of environmental uncertainty.

### C.7.2 Impact of LLM Uncertainty

To evaluate the impact of LLM uncertainty, we evaluate the impact of using different prompts and the impact of using different LLM temperatures.

**LLM Uncertainty: Changing Prompts**

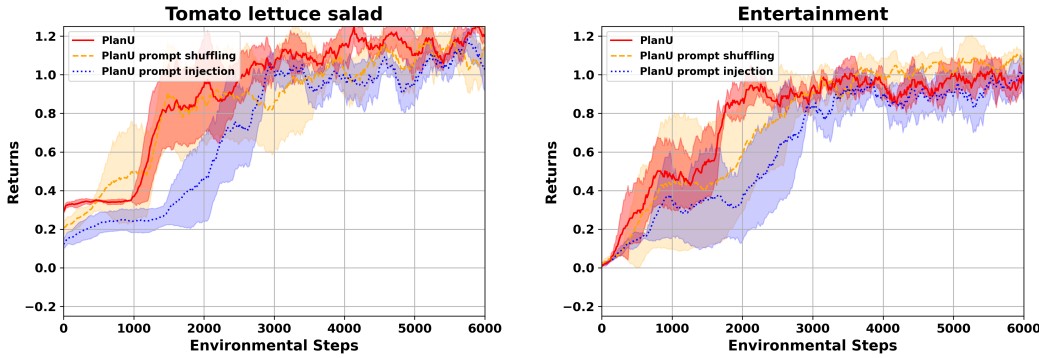

Figure 15: The impact of LLM uncertainty caused by changing prompts.

To evaluate our method's robustness to LLM uncertainty, we introduce two types of perturbations to LLM prompts, which lead to different outputs from LLMs given the same state: (1) **Prompt**

**Shuffling**: We shuffle the sentences in the provided prompt while preserving the sentence boundaries within the task and environment description. (2) **Prompt Injection**: We inject task-irrelevant but environment-related information into the prompt that does not aid in solving the task. These forms of uncertainty commonly arise when prompts include content generated by LLMs. Examples of such prompts and results for the *Tomato Lettuce Salad* task are provided in Appendix D. The experimental results are shown in Figure 15. As it is shown in the picture, the performance of PlanU does not significantly affected by the changed prompts. These results demonstrate that LLM uncertainty only slightly affects the convergence speed of PlanU. PlanU is robust to LLM uncertainty caused by changed prompts.

### LLM uncertainty: Generation Temperatures

To assess the robustness of planning strategies against the inherent uncertainty of large language models (LLMs), we examine the effect of varying the temperature parameter—a key factor influencing output stochasticity—on task success rates. The temperature $T$ modulates the softmax distribution over token logits $z_i$ as follows:

$$P(x_i) = \frac{\exp(z_i/T)}{\sum_j \exp(z_j/T)}, \tag{7}$$

where higher values of $T$ lead to flatter distributions, increasing randomness in generated outputs.

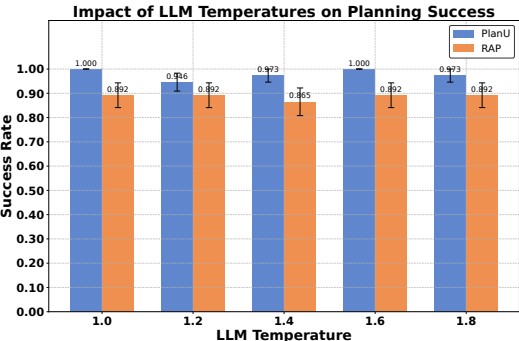

Figure 16: LLM Temperature for PlanU and RAP

As illustrated in Figure 16, PlanU exhibits a high degree of resilience to temperature variation. Across a range of temperature settings, the success rates remain largely stable, indicating that both methods are capable of maintaining consistent performance despite increasing randomness in LLM generations. PlanU consistently outperforms RAP under all temperature settings with shows lower variance, demonstrating its stronger robustness in uncertain generation environments.

# D   Prompts for LLM

In this section, we describe the prompt we use for LLM-based methods for the Blocksworld, Over-cooked(Fig 17) and Virtualhome(Fig 18).

**Overcooked and Virtualhome.** For both environments, we basically follow the experimental settings of [7], with additional uncertainty introduced in the prompts to evaluate our methods' capability to model such uncertainty.

---

**EXAMPLE PROMPT FOR OVERCOOKED**

```
Basic Prompt:
There is a fixed cutting board in the room.  Currently you don't
have anything in hand.  You notice a tomato on the table.  Your
goal is to serve the dish of a bowl only containing chopped
tomato.  Your next step is to

Prompt After shuffling:
Currently you don't have anything in hand.  Your goal is to serve
the dish of a bowl only containing chopped tomato.  There is
a fixed cutting board in the room.  You notice a tomato on the
table.  Your next step is to

Prompt after injection:
There are two fixed cutting boards in the room.  Earlier this day,
you made a dish with chopped potatoes.  Currently you don't have
anything in hand.  You notice a tomato on the table.  Your goal is
to serve the dish of a bowl only containing chopped tomato.  Your
next step is to
```

Figure 17: Example Prompt for Overcooked

---

**EXAMPLE PROMPT FOR OVERCOOKED**

```
Basic Prompt:
There are four rooms:  the kitchen, bathroom, bedroom, and living
room.  You are in the living room and you notice a coffee table, a
TV and a sofa.  Currently, you are not grabbing anything in hand.
Your goal is to enjoy the chips and the milk while watching TV.
Your next step is to

Prompt after shuffling:
Your goal is to enjoy the chips and the milk while watching TV.
There are four rooms:  the kitchen, bathroom, bedroom, and living
room.  Your goal is to enjoy the chips and the milk while watching
TV. You are in the living room and you notice a coffee table, a TV
and a sofa.  Your next step is to

Prompt after injection:
There are four rooms:  the kitchen, bathroom, bedroom, and living
room.  Earlier in the day, you were in the bedroom and sowe
cookies on the table.  You are in the living room and you notice
a coffee table, a TV and a sofa.  Currently, you are not grabbing
anything in hand.  Your goal is to enjoy the chips and the milk
while watching TV. Your next step is to
```

Figure 18: Example Prompt for Virtualhome.

## EXAMPLE DeLLMa PROMPT FOR BLOCKSWORLD

Prompt 1
I am interacting with a Blocksworld environment.  I need to make
optimal decisions about which action to perform next (e.g., pick
up, stack, unstack, or put down a block).
My goal is to place the blue block on top of the orange block.
Today's environment is partially observable, and I must reason
under uncertainty.

Here is the current observation:
- The blue block is clear.
- The orange block is clear.
- The hand is empty.
- The blue block is on the red block.
- The red block and the orange block are on the table.

The full world state is hidden and contains 16 latent variables
(e.g., hand status, block locations, perception errors, etc.).
*First, think about the unknown factors in the current environment
that could affect my decision.*

Prompt 2
*<SAME CONTEXT>*
Now I have enumerated the unknown variables that affect my
decision:
"block visibility", "block on block stability", "perception
error", "hand grip strength", ...

*Given these unknowns, think about the belief distribution over
their most likely values, with verbal confidences such as:  "very
likely", "likely", etc.*
Example format:
{"block visibility":  {"blue":  "very likely", "orange":  "very
likely"}, ...}

Prompt 3
*<SAME CONTEXT>*
Now, given the belief distribution over the environment state and
the goal ("blue block on orange block"), devise a step-by-step
plan.
You may use the following actions:
(1) pick up a block
(2) unstack a block from on top of another block
(3) put down a block
(4) stack a block on top of another block

Output one action per line.  End the plan with [PLAN END].

Example Output:
unstack the blue block from on top of the red block
stack the blue block on top of the orange block
[PLAN END]

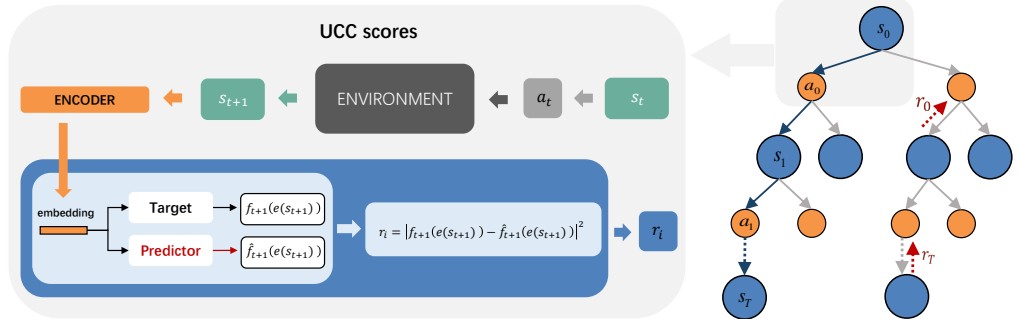

Figure 19: The Schematic Plot for the UCC score

# E   UCC Score

During tree search, PlanU chooses optimism in the face of uncertainty; it encourages exploration through the Upper Confidence Bound with Curiosity (UCC) score. It calculates the uncertainty of text-based states based on the value distribution and novelty of states. The schematic plot for the selection phase using the UCC score is depicted in Figure 19. To estimate the novelty of a given state, we first encode the natural language description of the state using a sentence-level encoder. Specifically, we utilize Sentence Transformers [3] to encode textual states into dense embeddings. These embeddings are fed into a fixed target network and a learnable predictor network, with their output discrepancy quantifying the state's novelty as a measure of epistemic uncertainty.

Specifically, the UCC score is defined as follows.

$$UCC(s_t, a_t) = \psi[Z(s_t, a_t)] + c_1 \cdot \frac{r_i(s_t)}{N(s_t, a_t)}, \tag{8}$$

where $\psi[Z(s_t, a_t)]$ is a uncertainty-aware operator that maps a quantile distribution into a real-value. Multiple implementations of $\psi$ can be used. In default, we use the expectation operator $\mathbb{E}$ as $\psi$. $c_1$ is a hyperparameter, $N(s_t, a_t)$ represents the visit count of the action node $a_t$, and $r_i(s_t)$ is a novelty reward.

The novelty reward $r_i(s_t)$ is designed to encourage the agent to explore less-visited states from a global perspective through estimating the uncertainty of $s_t$. Inspired by [17], $r_i(s_t)$ calculates the feature difference among a predictor network $\hat{f}$ and a target network $f$. Specifically, it is defined as $r_i(s_t) = \left| \hat{f}(e(s_t)) - f(e(s_t)) \right|^2$, where $\hat{f}$ and $f$ are both three-layer Multi-Layer Perceptron (MLP) networks.

Following [17], the hyperparameters used for the predictor network $\hat{f}$ are listed in Table 11. We use $r_i$ as the loss to train the predictor network.

Table 11: Hyperparameter Settings for UCC

| Hyper-parameter | Value |
|---|---|
| learning_rate | 1e-5 |
| hidden_size_list | [64, 64, 128] |
| update_per_collect | 5 |
| obs_norm | True |
| obs_norm_clamp_min | -1 |
| obs_norm_clamp_max | 1 |
| intrinsic_reward_weight | 0.01 |

---

[3]https://huggingface.co/sentence-transformers/all-mpnet-base-v2

# F Broader Impact and Limitation

The goal of PlanU is to advance the field of Machine Learning technique for Decision Making. Our work can be applied in multiple domains. However, we do not feel any thing must be highlighted here.

There are two limitations of PlanU. Firstly, PlanU does not use another tool (e.g., web search) during decision-making. This may limit its ability for problems requiring external knowledge. In such cases, a web search could help make the decision. We plan to combine PlanU and ReAcT to strengthen its ability. Secondly, PlanU is an MCTS-based decision-making method. It faces challenges when applied to problems with high-dimensional action. PlanU may struggle to explore sufficient paths with practical time constraints, leading to suboptimal performance. We plan to explore hierarchical MCTS, which decomposes the problem into hierarchical layers or combine PlanU with progressive widening techniques, which dynamically limit the number of actions explored per node.

