# OpenReview forum: "PlanU: Large Language Model Reasoning through Planning under Uncertainty"
_NeurIPS.cc/2025/Conference — NeurIPS 2025 poster_

### Official Review · Reviewer_6Yh1 · 2025-06-29

**Clarity:** 3
**Significance:** 3
**Originality:** 3
**Rating:** 4
**Confidence:** 4

**Summary:**

This paper proposes PlanU, an LLM-based planning framework that addresses both LLM and environmental uncertainty (e.g., stochastic state transitions). It combines quantile distribution modeling of return values and a UCC (Upper Confidence Bounds with Curiosity) exploration strategy within MCTS. The approach is evaluated on multiple benchmarks, including Blocksworld, Overcooked, and VirtualHome, using several LLM backbones. Experimental results show that PlanU consistently outperforms strong baselines.

**Questions:**

•  See the weakness section.
•  line 212: a uncertainty-aware - an uncertainty-aware

**Ethical Concerns:**

["NO or VERY MINOR ethics concerns only"]

**Final Justification:**

After considering the rebuttal and the discussions with the authors and other reviewers, my recommended score is borderline accept. My main concern was the lack of thorough experiments, which has been largely addressed.

**Limitations:**

yes

**Quality:**

3

**Strengths And Weaknesses:**

Strengths
• PlanU is the first to introduce quantile-based value estimation into LLM-driven MCTS, which is a notable step toward enabling more robust and uncertainty-aware planning in real-world LLM agents.
• Thorough testing across multiple benchmarks. The inclusion of variations such as prompt injection and prompt shuffling further strengthens the experimental credibility.

Weaknesses
• All tested backbones are in the 7–8B parameter range. Evaluating across more diverse model scales (e.g., smaller or larger LLMs) would improve the generality of the claims.
• The ablation study could be more fine-grained. For example, testing hybrid configurations such as quantile + standard UCT or mean value + UCC would better isolate the individual contributions of each module.

---

> ### Author Rebuttal · Authors · 2025-07-31
>
> > All tested backbones are in the 7–8B parameter range. Evaluating across more diverse model scales (e.g., smaller or larger LLMs) would improve the generality of the claims.
>
> We further evaluated larger models in Blocksworld, including Qwen2.5-14B-Instruct, OpenAI GPT-4.1, and Google Gemini-2.5-Pro. These cover both larger-scale open-source and closed-source models. The results are summarized in the following table. PlanU performs well both on larger-scale open-source and closed-source models.
>
> **OpenAI/GPT-4.1**
> | Method | 2-step          | 4-step          | 6-step          | 8-step          |
> |--------|-----------------|-----------------|-----------------|-----------------|
> | CoT    | 0.541 ± 0.07    | 0.382 ± 0.06    | 0.221 ± 0.05    | 0.063 ± 0.04    |
> | ToT    | 0.595 ± 0.08    | 0.421 ± 0.07    | 0.241 ± 0.06    | 0.077 ± 0.05    |
> | RAP    | 1.000 ± 0.00    | 0.763 ± 0.05    | 0.341 ± 0.03    | 0.231 ± 0.06    |
> | PlanU  | 1.000 ± 0.00    | 0.821 ± 0.04    | 0.648 ± 0.03    | 0.392 ± 0.04    |
>
> **Google/Gemini-2.5-Pro**
> | Method | 2-step          | 4-step          | 6-step          | 8-step          |
> |--------|-----------------|-----------------|-----------------|-----------------|
> | CoT    | 0.568 ± 0.08    | 0.395 ± 0.07    | 0.228 ± 0.06    | 0.056 ± 0.05    |
> | ToT    | 0.622 ± 0.07    | 0.408 ± 0.08    | 0.248 ± 0.07    | 0.070 ± 0.06    |
> | RAP    | 1.000 ± 0.00    | 0.776 ± 0.03    | 0.328 ± 0.04    | 0.238 ± 0.05    |
> | PlanU  | 1.000 ± 0.00    | 0.808 ± 0.05    | 0.662 ± 0.03    | 0.378 ± 0.04    |
>
> **Qwen/Qwen2.5-14B-Instruct**
> | Method | 2-step          | 4-step          | 6-step          | 8-step          |
> |--------|-----------------|-----------------|-----------------|-----------------|
> | CoT    | 0.405 ± 0.06     | 0.178 ± 0.05     | 0.138 ± 0.04     | 0.028 ± 0.03     |
> | ToT    | 0.486 ± 0.05     | 0.247 ± 0.06     | 0.131 ± 0.05     | 0.042 ± 0.04     |
> | RAP    | 0.973 ± 0.02     | 0.605 ± 0.03     | 0.219 ± 0.03     | 0.161 ± 0.05     |
> | PlanU  | 1.000 ± 0.00     | 0.750 ± 0.02     | 0.521 ± 0.05     | 0.182 ± 0.04     |
>
>
> > The ablation study could be more fine-grained. For example, testing hybrid configurations such as quantile + standard UCT or mean value + UCC would better isolate the individual contributions of each module.
>
> Thanks for your valuable advice. We have conducted such fine-grained ablation studies. Specifically, the hybrid configurations suggested are clearly presented in Appendix Figure 10:
> 1. For the mean value + UCC, it is depicted as 'PlanU w/o dist'.
> 2. The quantile + standard UCT is depicted as 'PlanU w/o UCC'.
>
> **Tomato lettuce salad**
> |                | 0              | 1000           | 2000           | 3000           | 4000          | 5000           | 6000           |
> | -------------- | -------------- | -------------- | -------------- | -------------- | ------------- | -------------- | -------------- |
> | PlanU          | 0.2941±0.0099  | 0.3696±0.0335  | 0.9644±0.2569  | 1.0568±0.0709  | 1.1509±0.1067 | 1.11±0.0978    | 1.2079±0.0324  |
> | PlanU w/o dist | 0.2134±0.0074  | 0.3339±0.0074  | 0.3358±0.003   | 0.3455±0.003   | 0.3383±0.0048 | 0.3441±0.0037  | 0.3477±0.0002  |
> | PlanU w/o UCC  | 0.1183±0.0     | 0.3081±0.0127  | 0.3326±0.0038  | 0.3339±0.0074  | 0.3371±0.0056 | 0.339±0.0067   | 0.7025±0.1744  |
>
> **Entertainment**
> |                | 0             | 1000          | 2000          | 3000          | 4000          | 5000          | 6000          |
> | -------------- | ------------- | ------------- | ------------- | ------------- | ------------- | ------------- | ------------- |
> | PlanU          | 0.0084±0.0044 | 0.46±0.2798   | 0.877±0.0911  | 0.9453±0.073  | 0.9487±0.0575 | 0.9914±0.0851 | 0.9822±0.0114 |
> | PlanU w/o dist | 0.0009±0.0    | 0.0701±0.0    | 0.085±0.0     | 0.1871±0.005  | 0.7159±0.1152 | 0.9062±0.0623 | 0.9779±0.0813 |
> | PlanU w/o UCC  | 0.0001±0.0074 | 0.1322±0.0314 | 0.1425±0.0356 | 0.562±0.2937  | 0.8316±0.2473 | 1.0291±0.1195 | 0.9509±0.1575 |
>
> Ablation results show PlanU outperforms PlanU w/o dist and PlanU w/o UCC in both scenarios, with faster convergence and better optimal performance—confirming the critical role of both components.
>
> To provide more information to the reviewer, we have conducted experiments to evaluate the impact of the text encoder used in UCC, and evaluate the impact of the number of quantiles.
>
> For text encoders, the default PlanU performs better in medium-to-high iterations across both scenarios. While the two all-MiniLM variants show slightly lower overall performance, their gap with the default narrows in later iterations, indicating UCC's compatibility with different encoders.
>
> **The different sentence-transformer models in Tomato salad**
> |                         | 0              | 1000          | 2000          | 3000          | 4000          | 5000          | 6000          |
> | ----------------------- | -------------- | ------------- | ------------- | ------------- | ------------- | ------------- | ------------- |
> | PlanU（default，all-mpnet-base-v2）        | -0.083±0.0415  | 0.8995±0.0832 | 1.1069±0.0032 | 1.096±0.0132  | 1.1045±0.0032 | 1.0955±0.0173 | 1.105±0.0039  |
> | PlanU_all-MiniLM-L6-v2  | -0.1089±0.0232 | 0.2991±0.0557 | 0.9031±0.0912 | 0.8501±0.1061 | 0.8632±0.1234 | 0.8748±0.0888 | 0.8627±0.1072 |
> | PlanU_all-MiniLM-L12-v2 | -0.0926±0.0317 | 0.1556±0.0282 | 0.8808±0.0938 | 0.8008±0.1062 | 0.8764±0.1031 | 0.8897±0.0898 | 0.8837±0.0896 |
>
> **The different sentence-transformer models in Tomato lettuce salad**
> |                         | 0             | 1000          | 2000          | 3000          | 4000          | 5000          | 6000          |
> | ----------------------- | ------------- | ------------- | ------------- | ------------- | ------------- | ------------- | ------------- |
> | PlanU（default）        | 0.2941±0.0099 | 0.3696±0.0335 | 0.9644±0.2569 | 1.0568±0.0709 | 1.1509±0.1067 | 1.11±0.0978   | 1.2079±0.0324 |
> | PlanU (all-MiniLM-L6-v2)  | 0.132±0.0531  | 0.6347±0.1836 | 0.7467±0.0918 | 0.9486±0.0978 | 0.7599±0.0919 | 0.9085±0.0928 | 0.9751±0.1106 |
> | PlanU (all-MiniLM-L12-v2) | 0.1802±0.0411 | 0.4936±0.13   | 0.7907±0.0819 | 0.8845±0.1031 | 1.1068±0.0956 | 0.8613±0.1147 | 1.0757±0.1077 |
>
> The default setting of 51 quantiles (n=51) outperforms both n=41 and n=61 in most iterations, particularly in later stages. Performance differences across quantile choices are minor, indicating that the method is robust to changes in this parameter, with the default providing the best results in our experiments.
>
> **The different number of quantiles in Tomato salad**
> |                  | 0              | 1000          | 2000          | 3000          | 4000          | 5000          | 6000          |
> | ---------------- | -------------- | ------------- | ------------- | ------------- | ------------- | ------------- | ------------- |
> | PlanU(default,n=51) | -0.0926±0.0317 | 0.1556±0.0282 | 0.8808±0.0938 | 0.8008±0.1062 | 0.8764±0.1031 | 0.8897±0.0898 | 0.8837±0.0896 |
> | PlanU(n=41)  | -0.0934±0.0305 | 0.2481±0.0407 | 0.8822±0.104  | 0.9154±0.0972 | 0.9159±0.0942 | 0.8274±0.0763 | 0.8577±0.1107 |
> | PlanU(n=61) | -0.0936±0.0262 | 0.2078±0.0573 | 0.9045±0.0887 | 0.9063±0.1    | 0.8844±0.0968 | 0.8028±0.0869 | 0.8631±0.0989 |
>
> **The different number of quantiles in Tomato lettuce salad**
> |                  | 0             | 1000          | 2000          | 3000          | 4000          | 5000          | 6000          |
> | ---------------- | ------------- | ------------- | ------------- | ------------- | ------------- | ------------- | ------------- |
> | PlanU (default, n=51) | 0.2941±0.0099 | 0.3696±0.0335 | 0.9644±0.2569 | 1.0568±0.0709 | 1.1509±0.1067 | 1.11±0.0978   | 1.2079±0.0324 |
> | PlanU (n=41)  | 0.1163±0.0539 | 0.5403±0.159  | 0.865±0.0882  | 0.8721±0.0956 | 1.04±0.1082   | 0.9891±0.0962 | 0.9802±0.1093 |
> | PlanU (n=61)  | 0.1656±0.0429 | 0.5706±0.1551 | 0.8318±0.1019 | 0.8276±0.1099 | 1.043±0.0982  | 0.9755±0.107  | 0.9901±0.1003 |
>
>
> >• line 212: a uncertainty-aware - an uncertainty-aware
>
> Thanks for your help. We will check the writing of the paper carefully by using a grammar checker.

---

### Official Review · Reviewer_QuAk · 2025-07-02

**Clarity:** 3
**Significance:** 2
**Originality:** 2
**Rating:** 2
**Confidence:** 3

**Summary:**

This paper proposes the PlanU framework, which combines the large language model (LLM) and tree search (MCTS) to find the best planning path in an uncertain environment. It uses distributed (rather than average) rewards to represent uncertainty and introduces curiosity to encourage exploration.

**Questions:**

1. In the setting of this paper, the introduction of uncertainty seems to be limited to the possibility that an action may fail. However, in most cases, wouldn’t simply re-executing the same action suffice? What is the intuitive benefit of introducing a reward distribution in this context? Moreover, given the vast optimization space, it raises concerns about what kind of distribution is actually being learned.

2. In Figure 5 (a), we observe that removing any single component results in almost no return improvement for the baseline. Why is this the case? Could the authors provide more explanation or insights? In Figure 5 (b) and (c), we see that the proposed method and the baseline do not seem to be clearly distinguishable, which reinforces my doubts about the validity of the experiment.

**Ethical Concerns:**

["NO or VERY MINOR ethics concerns only"]

**Limitations:**

Yes

**Quality:**

2

**Strengths And Weaknesses:**

**Strength**
1. The current mainstream exploration is indeed mainly concentrated in deterministic environments. LLM decision-making in uncertain environments is realistic and meaningful.

2. The experiments in this article prove the effectiveness of PlanU, which is more effective than other MCTS and RL baselines.

**Weakness**
1. The main technical contributions of this paper largely stem from prior works, such as Monte Carlo Tree Search (MCTS), Quantile Regression DQN (QRDQN) [1], and using LLMs as action priors. The novelty of estimating novelty via the discrepancy between two neural networks also follows the idea proposed in [2].

2. The experimental setup appears to be overly toy-like. Most of the failure cases are manually injected, which may not reflect the challenges of real-world environments.

3. In Equation (6), the authors mention that they use the expectation operator \( \mathbb{E} \) as the default \( \psi \). This raises strong concerns about the necessity of learning the full distribution \( Z(s_t, a_t) \), as this essentially reduces to using the mean reward. In my view, this significantly undermines the novelty of the method.

4. The proposed UCC score lacks theoretical justification, whereas classical UCT formulations offer well-established theoretical guarantees.

5. I believe the proposed framework does not truly exploit the capability of LLMs as agents. The LLM is only used for initializing probabilities. A more powerful and effective usage would be to adopt LLMs as ReAct-style agents that can dynamically reason and decide the next action based on environmental feedback.

**References:**

[1] Dabney, Will, et al. "Distributional reinforcement learning with quantile regression." *Proceedings of the AAAI Conference on Artificial Intelligence*. Vol. 32. No. 1. 2018.

[2] Burda, Yuri, et al. "Exploration by random network distillation." *arXiv preprint arXiv:1810.12894* (2018).

[3] Yao, Shunyu, et al. "ReAct: Synergizing reasoning and acting in language models." *International Conference on Learning Representations (ICLR)*. 2023.

---

> ### Author Rebuttal · Authors · 2025-07-31
>
> > W1:The main technical contributions of this paper largely stem from prior works, such as MCTS, QRDQN...
>
> Regarding the technical contribution and novelty, we hold a different opinion from the reviewer.
>
> 1. It is common to address a problem through the integration of existing approaches. DQN is an integration of Q-Learning with a deep neural network. Self-consistency is a combination of CoT with majority voting. ToT extends CoT with a tree structure. RAP combines MCTS with LLM. Rainbow combines double Q-learning, prioritized replay, dueling networks, multi-step learning, distributional RL, and noisy nets. These works make a significant contribution to research advancement.
> 2. Our contribution is threefold. We are **the first to identify the common uncertainty pitfall when using vanilla MCTS for LLM planning**. Second, we are the first to integrate the quantile distribution into MCTS-based LLM planning. Third, we are the first to integrate random network distillation into MCTS-based LLM planning.
> 3. **Addressing the uncertainty pitfall when using MCTS for LLM planning is non-trivial**, none of the existing LLM-based planning approaches addresses the environmental uncertainty and the LLM uncertainty together. We address the uncertainty pitfall through **a novel integration of existing approaches**. We had shown in Figure 3 that adding uncertainty consideration in CoT, DeLLMa, RAP, and Reflextion fail for stochastic environments.
> 4. We have conducted multiple experiments combining existing uncertainty-aware methods into LLM planning. RAP-D and RAP-E combine RAP with DMCTS and EMCTS, respectively. DMCTS and EMCTS are methods dealing with uncertainty. RAP-D-UCC replaces the UCT used in RAP-D with the UCC score. As it is shown in the following tables, **simple integrations of existing methods that deal with uncertainty do not work effectively**.
>
> **Tomato salad**
> ||0|1000|2000|3000|4000|5000|6000|
> |-|-|-|-|-|-|-|-|
> |PlanU|0.083±0.041|0.8995±0.083|1.1069±0.003|1.096±0.013|1.104±0.003|1.095±0.017|1.105±0.003|
> |RAP(LLM+UCT)|0.051±0.012|0.367±0.072|0.475±0.067|0.268±0.073|0.679±0.061|0.584±0.064|0.258±0.080|
> |RAP(LLM+pUCT)|0.064±0.058|0.014±0.087|0.989±0.075|0.941±0.066|0.967±0.053|0.987±0.058|0.960±0.045|
> |RAP-D (DMCTS+pUCT)|0.158±0.00|0.128±0.006|0.139±0.006|0.149±0.003|0.153±0.003|0.149±0.007|0.151±0.004|
> |RAP-E (EMCTS+pUCT)|0.104±0.047|0.118±0.023|0.357±0.059|0.393±0.109|0.346±0.118|0.465±0.113|0.319±0.113|
> |RAP-D-UCC|0.044±0.076|0.691±0.188|0.886±0.277|0.898±0.229|0.875±0.292|0.882±0.254|0.896±0.225|
>
> **Tomato lettuce salad**
> ||0|1000|2000|3000|4000|5000|6000|
> |-|-|-|-|-|-|-|-|
> |PlanU|0.294±0.009|0.369±0.033|0.964±0.256|1.056±0.070|1.150±0.106|1.11±0.097|1.207±0.032|
> |RAP(LLM+UCT)|-0.031±0.026|-0.037±0.036|0.318±0.084|0.245±0.082|0.267±0.081|0.308±0.078|0.274±0.078|
> |RAP(LLM+pUCT)|0.139±0.031|0.296±0.034|0.318±0.01|0.320±0.017|0.318±0.004|0.327±0.010|0.324±0.012|
> |RAP-D (DMCTS+pUCT)|0.030±0.004|0.110±0.019|0.140±0.01|0.142±0.011|0.142±0.009|0.19±0.016|0.261±0.018|
> |RAP-E (EMCTS+pUCT)|0.104±0.0|0.252±0.065|0.533±0.282|0.603±0.34|0.562±0.279|0.521±0.24|0.508±0.244|
> |RAP-D-UCC|0.116±0.007|0.284±0.039|0.334±0.006|0.971±0.092|0.952±0.118|0.991±0.105|1.065±0.132|
>
> > W2:...Most of the failure cases are manually injected..
>
> We have evaluated PlanU on real-world tasks with more realistic failures. Please refer to our response to Reviewer c3qH and bLkk.
>
> > W3:...the expectation operator ( \mathbb{E} ) as the default ( \psi )...the necessity of learning the full distribution ( Z(s_t, a_t) )...
>
> In distributional RL, **learning an approximate distribution rather than a single expected value can preserve the multimodality in value distribution, which can lead to more stable learning.** As shown in the following tables, PlanU w/o dist—a variant using mean value without learning quantile distribution—performs significantly worse than PlanU.
>
> We also evaluated impacts of different operators: $\tau_ {0.5}$, $\mathbb{E}[Z(s_ t,a_ t)]$ +variance, and $\mathbb{E}[Z(s_ t,a_ t)]$+ $\tau_ {0.9} - \tau_ {0.1}$. $\mathbb{E}[Z(s_ t,a_ t)]$ is the mean operator, $\tau_ {0.5}$, $\tau_ {0.9}$ and $\tau_ {0.1}$ are the value for quantile 0.5, 0.9 and 0.1, respectively. The mean operator (below tables) can lead to first or the second performance.
>
> **Tomato lettuce salad**
> ||0|1000|2000|3000|4000|5000|6000|
> |-|-|-|-|-|-|-|-|
> |PlanU|0.294±0.009|0.369±0.033|0.964±0.256|1.056±0.070|1.150±0.106|1.11±0.097|1.207±0.032|
> |PlanU w/o dist|0.213±0.007|0.333±0.007|0.335±0.003|0.345±0.003|0.338±0.004|0.344±0.003|0.347±0.001|
> |$\tau_{0.5}$|0.051±0.046|0.524±0.074|0.567±0.077|0.729±0.105|0.765±0.09|0.626±0.101|1.027±0.092|
> |$\mathbb{E}[Z(s_ t,a_ t)]$+variance|0.102±0.058|0.192±0.069|0.234±0.056|0.210±0.059|0.867±0.113|0.966±0.115|0.922±0.088|
> |$\mathbb{E}[Z(s_ t,a_ t)]$+$\tau_ {0.9}-\tau_ {0.1}$|0.090±0.073|0.916±0.102|0.929±0.087|0.889±0.098|1.022±0.108|0.905±0.097|1.067±0.089|
>
>  **Entertainment**
> ||0|1000|2000|3000|4000|5000|6000|
> |-|-|-|-|-|-|-|-|
> |PlanU|0.008±0.004|0.46±0.279|0.877±0.091|0.945±0.073|0.948±0.057|0.991±0.085|0.982±0.011|
> |PlanU w/o dist|0.001±0.0|0.070±0.0|0.085±0.0|0.1871±0.005|0.7159±0.115|0.906±0.062|0.977±0.081|
> |$\tau_ {0.5}$|0.004±0.007|0.22±0.009|0.234±0.004|0.849±0.24|0.858±0.239|0.804±0.018|0.983±0.024|
> |$\mathbb{E}[Z(s_ t,a_ t)]$+variance|-0.004±0.01|0.0607±0.009|0.293±0.079|0.773±0.239|0.712±0.239|0.932±0.021|0.975±0.031|
> |$\mathbb{E}[Z(s_ t,a_ t)]$+$\tau_ {0.9}-\tau_ {0.1}$|-0.016±0.001|0.033±0.005|0.485±0.112|0.785±0.223|0.887±0.077|1.012±0.007|0.940±0.059|
>
> > W5:...The LLM is only used for initializing probabilities... ReAct-style agents...  based on environmental feedback.
>
> 1. As stated (Lines 29-32), PlanU leverages LLMs beyond simple initialization: it explicitly uses **them as a world model** to simulate outcomes for planning, similar to humans predicting consequences before execution.
>
> 2. PlanU **does incorporate environmental feedback**. As detailed in Figure 2 and Lines 199-206, our backpropagation mechanism dynamically updates action probabilities based on real-world outcomes.
>
> 3. In line with the reviewer's suggestion, **we did compare our work with powerful agents** in Figure 3 and Appendix C2.2: a ReAct-style MCTS agent (LATS, ICML 24) and an uncertainty-aware agent (DeLLMa, ICLR 25). LATS uses ReAct agent and reflection during MCTS planning. DeLLMa infers and predicts hidden factors to make decision. The results (below table) show that PlanU performs better than such powerful agents.
>
> |Method|2-step|4-step|6-step|8-step|
> |-|-|-|-|-|
> |LATS|0.946±0.037|0.684±0.053|0.352±0.04|0.077±0.023
> |DeLLMa|0.595±0.081|0.092±0.033|0.041±0.017|0.000±0.000|
> |PlanU|1.000±0.0|0.808±0.05|0.662±0.03|0.378±0.04|
>
> > Q1:...wouldn’t simply re-executing the same action suffice? What is the intuitive benefit of introducing a reward distribution...
>
> If an action is optimal, re-executing it suffices, but the agent must learn to know which action is the optimal. Thus, introducing a quantile distribution is necessary to better model potential multimodality in the value distribution. We have explored more realistic failure models. Please refer to our response to Reviewer c3qH and bLkk.
>
> > Q2:...removing any single component results in almost no return improvement for the baseline...the proposed method and the baseline do not seem to be clearly distinguishable...
>
> Sorry for the misunderstanding.
>
> 1. There are not results for any baselines in Figures 5 (a) and (b). For comparison, we additionally show the results of RAP in below tables. **The results show that each component of PlanU leads to performance improvement thanks to their ability to deal with uncertainty.**
>
> **Tomato lettuce salad**
> ||0|1000|2000|3000|4000|5000|6000|
> |-|-|-|-|-|-|-|-|
> |PlanU|0.294±0.009|0.369±0.033|0.964±0.256|1.056±0.070|1.150±0.106|1.11±0.097|1.207±0.032|
> |PlanU w/o dist|0.213±0.007|0.333±0.007|0.335±0.003|0.345±0.003|0.338±0.004|0.344±0.003|0.347±0.001|
> |PlanU w/o ucc|0.1183±0.0|0.3081±0.012|0.332±0.003|0.333±0.007|0.337±0.005|0.339±0.006|0.702±0.174|
> |RAP|0.139±0.031|0.296±0.034|0.318±0.01|0.320±0.017|0.318±0.004|0.327±0.01|0.324±0.012|
>
> **Entertainment**
> ||0|1000|2000|3000|4000|5000|6000|
> |-|-|-|-|-|-|-|-|
> |PlanU|0.008±0.004|0.46±0.28|0.877±0.091|0.945±0.073|0.949±0.058|0.991±0.085|0.982±0.011|
> |PlanU w/o dist|0.001±0.0|0.07±0.0|0.085±0.0|0.187±0.005|0.716±0.115|0.906±0.062|0.978±0.081|
> |PlanU w/o ucc|0.000±0.007|0.132±0.031|0.143±0.036|0.562±0.294|0.832±0.247|1.029±0.12|0.951±0.158|
> |RAP|-0.004±0.0|0.070±0.0|0.094±0.0|0.542±0.232|0.896±0.045|0.728±0.012|0.828±0.022|
>
> 2. For Figure 5c, **we demonstrate that PlanU deals with LLM uncertainty well, even for ambiguous instructions**. To augment Figure 5c, we evaluated RAP under prompt shuffling (randomly shuffling task/environment description sentences) and injection (adding task-irrelevant but environment-related text)—uncertainties common when prompts include LLM-generated content. Results (below table) show PlanU consistently outperforms RAP in handling LLM uncertainty.
>
> **Entertainment**
> ||0|1000|2000|3000|4000|5000|6000|
> |-|-|-|-|-|-|-|-|
> |PlanU|0.008±0.004|0.460±0.279|0.877±0.091|0.945±0.073|0.949±0.058|0.991±0.085|0.982±0.011|
> |PlanU prompt shuffling|0.0138±0.011|0.442±0.345|0.602±0.310|0.906±0.090|0.986±0.0997|1.017±0.060|1.065±0.046|
> |PlanU prompt injection|0.011±0.006|0.341±0.179|0.376±0.232|0.839±0.042|0.888±0.048|0.898±0.106|0.997±0.088|
> |RAP|-0.004±0.0|0.07±0.0|0.094±0.0|0.542±0.232|0.896±0.045|0.728±0.012|0.828±0.022|
> |RAP prompt shuffling|0.077±0.037|0.058±0.044|0.059±0.038|0.087±0.025|0.094±0.029|0.482±0.041|0.464±0.027|
> |RAP prompt injection|0.039±0.031|0.071±0.022|0.104±0.017|0.172±0.011|0.144±0.036|0.212±0.039|0.261±0.05|
>
> > W4:The proposed UCC score lacks theoretical justification...
>
> UCT converges to optimal actions for the tabular case with finite states. However, it is not guaranteed that UCT will converge for neural networks. We leave the theoretical guarantees of UCC as future work.

---

> ### Author Response · Authors · 2025-08-04
>
> Dear Reviewer QuAk,
>
> Thanks for your effort in reviewing our work. We deeply appreciate your time and suggestions. NeurIPS is renowned for its rigorous review process. Throughout this process, reviewers provide valuable feedback on both the manuscript and author responses, which significantly fosters progress within the research community. **To help us address your concerns most effectively, could you kindly specify any remaining points requiring further clarification?**
>
> We have addressed your concerns in the rebuttal. We summarize our response as follows:
>
> 1. We addressed your concerns regarding **technical novelty and effectiveness** (W1) through comparative experiments and ablations, including variants of RAP with uncertainty modeling, as well as comparisons with strong agent (ReAct-style) baselines such as LATS (ICML 24) and DeLLMa (ICLR 25). The results demonstrate that simple combinations of existing methods fail in uncertain settings, while PlanU performs robustly across tasks.
> 2. We addressed concerens about the **experiments(W2)** by adding two real-world benchmarks: TravelPlanner and Webshop.
> 3. We addressed concerns about the **value of modeling reward distributions** (W3, Q1) by showing that PlanU outperforms its mean-value variant (PlanU w/o dist), and that distributional modeling enables better handling of multimodal returns.
> 4. We have described the difference among UCT and UCC (W4).
> 5. We clarified that PlanU uses LLMs not only for initialization but also as **world models** and incorporates **environmental feedback** during MCTS updates (W5). PlanU thus enables more dynamic and informed planning than methods that treat LLMs statically.
> 6. We addressed concerns about component effectiveness and baseline distinguishability (Q2) by clarifying the results of Figures 5(a)(b) with baseline RAP and further evaluate them under prompt perturbations.
>
>
> ---
>
> **In summary, based on your feedback we added:**
>
> 1. Evaluating PlanU and others on two new benchmarks: WebShop and TravelPlanner. In WebShop, the agent needs to buy products according to user instructions, with real internet latency injected to simulate uncertainty. In TravelPlanner, the agent must plan transportation, accommodation, and attractions according to the user's instructions. Real-world transportation delays are injected to simulate uncertainty.
> 3. Evaluating robustness to **prompt perturbations**, including shuffled or injected instructions.
> 4. Direct **comparisons with ReAct-style and uncertainty-aware agents:** LATS (ICML 24）and DeLLMa (ICLR 25).
> 5. Additional **ablations** to quantify the contribution of each PlanU component (e.g., distributional value estimates, UCC).
>
>
>
> **Do you have any remaining concerns? We are eager to discuss them with you.**
>
> Best Regards,
> Authors

---

> ### Author Response · Authors · 2025-08-05
> **Supplementary Proofs for the UCC Score (Part 1)**
>
> We show that the UCC score converges to the expectation of the quantile distribution under a tabular setting, if the expectation operator is used. Selecting the action according to the argmax operator on the UCC score can lead to maximizing expected returns. For this, we first prove Lemma regarding the convergence of the novelty score, and then prove the Theorem regarding the convergence of the UCC score.
>
> **Lemma 1** (Convergence of Novelty Score). Consider a finite state space $\mathcal{S}$, an encoder $e: \mathcal{S} \to \mathcal{Z}$, and a fixed target function $f: \mathcal{Z} \to \mathbb{R}^k$. Let $\hat{f}: \mathcal{Z} \to \mathbb{R}^k$ be a predictor implemented as a tabular function over the finite latent space $\mathcal{Z}' = e(\mathcal{S})$ (i.e., $\hat{f}(z)$ is an independent learnable vector for each $z \in \mathcal{Z}'$). The novelty reward is defined by the following:
> \begin{align}
> 	r_i(s_t) = \|\hat{f}(e(s_t)) - f(e(s_t))\|_2^2.
> \end{align}
>
> Under the following assumptions:
>
> 1. Infinite visitation: Every state $s \in \mathcal{S}$ is visited infinitely often.
>
> 2. Robbins-Monro conditions: For each $z \in \mathcal{Z}'$, the learning rates $\{\alpha_n^{(z)}\}_ {n=1}^\infty$ for updates to $\hat{f}(z)$ satisfy:
> \begin{align}
> 	\sum_ {n=1}^\infty \alpha_ n^{(z)} = \infty \quad \text{and} \quad w\sum_ {n=1}^\infty \left(\alpha_ n^{(z)}\right)^2 < \infty.
> \end{align}
>
> Thus, the novelty score almost surely converges to zero:
> \begin{align}
> 	\lim_{t \to \infty} r_i(s_t) = 0 \quad \text{a.s.}
> \end{align}
>
> **Proof of Lemma 1.**
>
> Since the $\mathcal{S}$ is finite, $\mathcal{Z}' = e(\mathcal{S})$ is also finite. Let $\mathcal{Z}' = \{z_1, \dots, z_M\}$ where $M \leq |\mathcal{S}|$. For any latent point $z \in \mathcal{Z}'$, let $\mathcal{S}_z = \{s \in \mathcal{S} : e(s) = z\}$. By assumption 1, every state $s \in \mathcal{S}_z$ is visited infinitely often. Consequently, each $z$ is updated infinitely often.
>
> At the $n$-th update to $z$, let $\hat{f}^{(n)}(z)$ be the predictor's value and $e_n(z) = \hat{f}^{(n)}(z) - f(z)$. The SGD update minimizes $\mathcal{L}(z) = \|e_ n(z)\|_ 2^2$ and the rule is given by:
> \begin{align}
> 	\hat{f}^{(n+1)}(z) = \hat{f}^{(n)}(z) - \alpha_n^{(z)} \cdot 2e_n(z),
> \end{align}
> This update rule implies the following recursive relationship for the error:
> \begin{align}
> 	e_{n+1}(z) = \left(1 - 2\alpha_n^{(z)}\right) e_n(z).
> \end{align}
> As the update is identical for each dimension of $e_n(z) \in \mathbb{R}^k$, for the $j$-th component $e_n^{(j)}(z)$, the recursion is:
> \begin{align}
> 	e_{n+1}^{(j)}(z) = \left(1 - 2\alpha_n^{(z)}\right) e_n^{(j)}(z).
> \end{align}
> By solving this recursion, we obtain an expression for the error at the $n$-th step:
> \begin{align}
> 	e_n^{(j)}(z) = e_1^{(j)}(z) \prod_{m=1}^{n-1} \left(1 - 2\alpha_m^{(z)}\right).
> \end{align}
>
> The Robbins-Monro condition, $\sum_m^\infty (\alpha_m^{(z)})^2 < \infty$, implies that $\alpha_m^{(z)} \to 0$ as $m \to \infty$. Thus, $\exists M_z$ such that $0 < 2\alpha_m^{(z)} < 1$ for all $m \geq M_z$. The series $\sum_{m=1}^{\infty} \log\left(1 - 2\alpha_m^{(z)}\right)$ diverges to $-\infty$ because:
> \begin{align}
> 	\log\left(1 - 2\alpha_m^{(z)}\right) \leq -2\alpha_m^{(z)} \quad \forall \alpha_m^{(z)} \in (0,1/2),
> \end{align}
> and the Robbins-Monro condition states that $\sum_{m=1}^{\infty} -2\alpha_m^{(z)} = -\infty$ by $\sum \alpha_m^{(z)} = \infty$. Thus, $\prod_{m=1}^{\infty} \left(1 - 2\alpha_m^{(z)}\right) = 0$.
>
> Finally, we consider any state $s_t \in \mathcal{S}$ and its latent representation $z_t = e(s_t)$. As $t \to \infty$, the number of updates to $z_t \to \infty$. Therefore, the error $e_n(z_t)$ converges to zero. For the novelty score, the following equation holds:
> \begin{align}
> 	\lim_{t \to \infty} r_i(s_t) = \lim_{n \to \infty} \|e_n(z_t)\|_2^2 = 0 \quad \text{a.s.}
> \end{align}
>
> The Lemma 1 is proven.

---

> ### Author Response · Authors · 2025-08-05
> **Supplementary Proofs for the UCC Score (Part 2)**
>
> **Theorem 1** (Convergence of UCC). Consider a finite MDP with state space $\mathcal{S}$, action space $\mathcal{A}$, and an encoder $e: \mathcal{S} \to \mathcal{Z}$. Let $f: \mathcal{Z} \to \mathbb{R}^k$ be a fixed target function and $\hat{f}: \mathcal{Z} \to \mathbb{R}^k$ be a predictor implemented as a tabular function over $\mathcal{Z}' = e(\mathcal{S})$. We define the novelty score as the prediction error $r_ i(s_ t) = \|\hat{f}(e(s_t)) - f(e(s_t))\|^2_2$. The UCC score is given by $\mathrm{UCC}(s_ t, a_ t) = \mathbb{E}[Z(s_ t, a_ t)] + c_1 \cdot \frac{r_ i(s_ t)}{N(s_ t, a_ t)}$, where $c_1 > 0$ and $N(s_t, a_t)$ is the visit count of $(s_ t, a_ t)$.
>
> Under the following assumptions:
>
> 1. Every state $s \in \mathcal{S}$ is visited infinitely often.
>
> 2. For each $z \in \mathcal{Z}'$, the learning rates $\{\alpha_n^{(z)}\}$ satisfy the Robbins-Monro conditions:
> \begin{align}
> 	\sum_{n=1}^\infty \alpha_n^{(z)} = \infty, \quad \sum_{n=1}^\infty (\alpha_n^{(z)})^2 < \infty.
> \end{align}
>
> Then, the UCC score almost surely converges to the expectation of the quantile distribution:
> \begin{align}
> 	\lim_{t \to \infty} \mathrm{UCC}(s_t, a_t) = \mathbb{E}[Z(s_t, a_t)] \quad \text{a.s.}
> \end{align}
>
> **Proof of Theorem 1**
>
> We first decompose the UCC score and obtain that:
> \begin{align}
> 	\mathrm{UCC}(s_t, a_t) - \mathbb{E}[Z(s_t, a_t)] = c_1 \cdot \frac{r_i(s_t)}{N(s_t, a_t)}.
> \end{align}
> From Lemma 1, $r_i(s_t) \to 0$ a.s. Furthermore, since counts start at 1, $N(s_t, a_t) \geq 1$ for all $t$. Thus, the following inequality holds:
> \begin{align}
> 	0 \leq \frac{r_i(s_t)}{N(s_t, a_t)} \leq r_i(s_t) \quad \forall t.
> \end{align}
> As $r_i(s_t) \to 0$ a.s., we can obtain the following result by the squeeze theorem:
> \begin{align}
> 	\lim_{t \to \infty} \frac{r_i(s_t)}{N(s_t, a_t)} = 0 \quad \text{a.s.}
> \end{align}
> Consequently, we conclude that the difference between the UCC score and the expectation of the quantile distribution converges to zero almost surely:
> \begin{align}
> 	\lim_{t \to \infty} \left( \mathrm{UCC}(s_t, a_t) - \mathbb{E}[Z(s_t, a_t)] \right) = 0 \quad \text{a.s.},
> \end{align}
> which implies $\mathrm{UCC}(s_t, a_t) \to \mathbb{E}[Z(s_t, a_t)]$ a.s.
>
> The Theorem 1 is proven.
>
> Therefore, we discovered a key insight. The term $\frac{r_i(s_t)}{N(s_t, a_t)}$ vanishes almost surely because:
> 1. $r_i(s_t) \to 0$ due to infinite state visitation and SGD convergence.
> 2. $N(s_t, a_t) \geq 1$ prevents divergence.
>
> The result holds for any encoder $e$ since $\mathcal{Z}'$ remains finite.

---

> > ### Author Response · Authors · 2025-08-06
> >
> > Dear Reviewer QuAk,
> >
> > Thank you again for your thoughtful review. We have summarized and addressed your concerns in the rebuttal, including new real-world benchmarks, additional ablations, direct comparisons with ReAct-style and uncertainty-aware agents, and a theoretical proof for the UCC score.
> >
> > Could you kindly let us know if there are any remaining points requiring clarification? We would be glad to provide further details if needed.
> >
> > Best regards,
> >
> > Authors

---

> > > ### Author Response · Authors · 2025-08-08
> > >
> > > Dear Reviewer QuAk,
> > >
> > > There is only 1 day remaining until the new deadline for author-reviewer discussions. We kindly ask you if there are any remaining concerns. If we have addressed all your concerns, can you please reconsider your rating of this work?
> > >
> > > Sincerely,
> > >
> > > Authors of PlanU

---

> > > > ### Author Response · Authors · 2025-08-09
> > > >
> > > > Dear Reviewer QuAk,
> > > >
> > > > Thank you for your effort in reviewing our work. The discussion will close in less than 8 hours. We hope our responses have fully addressed your concerns; if so, could you please reconsider your rating?
> > > >
> > > > Best regards,
> > > >
> > > > Authors of PlanU

---

> ### Author Response · Authors · 2025-08-09
>
> Dear Reviewer QuAk,
>
> Thank you for your effort in reviewing our work. The author-reviewer discussion phase will be closed in **less than 4 hours**. We hope our responses have fully addressed your concerns; if so, could you please reconsider your rating? Are there any remaining concerns?
>
> Best regards,
>
> Authors of PlanU

---

### Official Review · Reviewer_bLkk · 2025-07-03

**Clarity:** 2
**Significance:** 2
**Originality:** 2
**Rating:** 4
**Confidence:** 4

**Summary:**

This paper introduces PlanU, a planning algorithm for LLM-based agents that integrates a distributional reinforcement learning approach into an MCTS framework. The core contribution is the representation of state-action values as a probability distribution rather than a scalar estimate. The authors argue that this allows the planner to more effectively handle decision-making in environments with stochastic state transitions. The method is evaluated on modified versions of some language-based planning tasks, where it is shown to outperform several baseline MCTS-based LLM planners.

**Questions:**

1. Related to the point discussed in weaknesses, canonical MCTS algorithms like UCT can handle stochastic environments. Could you please clarify why they are insufficient for the problems you are tackling, thereby requiring a distributional value function? Could you for example directly extend an UCT algorithm to LLMs and use it as a baseline?

2. Could you justify the specific choice of initializing the value distribution Z based on the policy's action probability during expansion? An ablation study on this initialization choice would be helpful.

**Ethical Concerns:**

["NO or VERY MINOR ethics concerns only"]

**Final Justification:**

I've raised my score to Borderline Accept based on the author rebuttal. The authors provided crucial new experiments that addressed the original submission's weak comparisons and evaluation. My recommendation is contingent on these results being fully integrated into the revised paper, which would make the work strong enough for acceptance.

**Limitations:**

I would suggest the authors discuss the following points:

* Generalization and scope: The evaluation is confined to planning tasks with artificially injected, simplistic stochasticity. The authors should discuss the challenges and potential performance implications of applying PlanU to domains with more complex and naturalistic uncertainty.

* Failure modes: A discussion of potential failure modes would be valuable.

**Quality:**

2

**Strengths And Weaknesses:**

__Strengths__

* Significance & originality: The paper addresses the important and challenging problem of enabling LLMs to plan robustly under uncertainty. The primary idea of incorporating a distributional value representation into an LLM-based planner is novel. In principle, a value distribution can carry more information than a scalar mean, potentially allowing for more sophisticated and risk-aware planning. This represents an interesting direction for research in LLM agents.

__Weaknesses__

* Flawed motivation and positioning: The paper's motivation is built on a questionable framing of the problem. The argument has two primary weaknesses:
  * Incorrect claims about MCTS: The paper repeatedly asserts that standard MCTS is unsuitable for stochastic environments (e.g., Lines 104, 161). This is factually incorrect. The canonical UCT algorithm was explicitly designed for and is proven to be asymptotically optimal in stochastic settings (MDPs).
  * Mischaracterizing a design choice as a fundamental limitation: While it may be true that the specific LLM-based MCTS baselines compared against in the paper were not designed for stochastic outcomes, this ---- in my opinion --- should be viewed as a simplifying implementation choice, likely because their original target tasks were deterministic. It is not, as this paper implies, a fundamental limitation of the MCTS framework itself when applied to LLMs.

* Quality of evaluation: The choice of benchmarks and the method of introducing stochasticity feel contrived. Introducing stochasticity by, for example, having a random chance of an action failing, feels artificial and does not necessarily represent the kinds of uncertainty LLM agents will face in more naturalistic settings (e.g., ambiguous instructions, partially observable states, etc). This choice of evaluation makes it difficult to assess the true generalizability and impact of the proposed method on problems with genuine, inherent uncertainty.

* The rationale for initializing the value distribution Z during the expansion phase using the policy's sampling probability is not justified. It is unclear why this is a sensible choice or how sensitive the algorithm's performance is to this specific initialization strategy.

---

> ### Author Rebuttal · Authors · 2025-07-31
>
> > W1: Incorrect claims about MCTS: ...standard MCTS is unsuitable for stochastic environments...canonical UCT algorithm was explicitly designed for...stochastic settings.
>
> Sorry for the misunderstanding about the motivation and positioning. We will make them clearer.
>
> 1. We agree with the reviewer that canonical UCT [1] guarantees convergence probability 100% for selecting optimal actions (i.e., action stochasticity $p(a|s)$), as established in the original UCT analysis. However, this action selection stochasticity ($p(a|s)$) is fundamentally distinct from environmental stochasticity ($p(s'|s,a)$) considered in this work. The former governs exploration/exploitation trade-offs during decision-making, while the latter characterizes transition uncertainty inherent to the environment itself.
>
> 2. The canonical UCT algorithm is unsuitable for stochastic environments, as discussed in [2].  We have shown in Figure 3 that UCT (written as LLM-MCTS) cannot model the true tree structure, which lead to wrong Q value estimation. Sparse UCT [2] addresses this drawback by allowing multiple child nodes per action (corresponding to different stochastic outcomes), while retaining the UCT action selection policy. Conceptually, Sparse UCT can be viewed as a simplified variant of our PlanU method, lacking its quantile distribution and UCC components. pUCT [3] improve planning through incorporating action priors (i.e., LLM priors). Our results (detailed below) demonstrate PlanU's superior performance compared to UCT, Sparse UCT, pUCT for LLM planning.
>
> **Tomato salad**
> ||0|1000|2000|3000|4000|5000|6000|
> |-----------------|-------------|-------------|-------------|-------------|-------------|-------------|-------------|
> |PlanU|0.2941±0.0099|0.3696±0.0335|0.9644±0.2569|1.0568±0.0709|1.1509±0.1067|1.11±0.0978|1.2079±0.0324|
> |RAP (LLM+UCT)|0.0516±0.0128|0.3675±0.0729|0.4756±0.0678|0.2687±0.0736|0.679±0.0618|0.5844±0.064|0.2588±0.0804|
> |LLM+Sparse UCT|0.1053±0.0469|0.2046±0.073|0.5546±0.0836|0.5415±0.0748|0.5491±0.0816|0.5522±0.0809|0.5453±0.076|
> |RAP(LLM+pUCT)|-0.0645±0.0588|-0.0143±0.0878|0.9895±0.0755|0.941±0.0662|0.9678±0.0533|0.9868±0.058|0.9603±0.0453|
>
>
> **Tomato lettuce salad**
> |                   | 0             | 1000          | 2000          | 3000          | 4000          | 5000          | 6000          |
> | ----------------- | ------------- | ------------- | ------------- | ------------- | ------------- | ------------- | ------------- |
> | PlanU             | -0.083±0.0415 | 0.8995±0.0832 | 1.1069±0.0032 | 1.096±0.0132  | 1.1045±0.0032 | 1.0955±0.0173 | 1.105±0.0039  |
> | RAP (LLM+UCT)      | -0.0315±0.0268 | -0.0374±0.0362 | 0.3184±0.0847 | 0.245±0.0826  | 0.2676±0.0812 | 0.3081±0.0786 | 0.2741±0.0786 |
> | LLM+Sparse UCT | 0.0142±0.0538 | 0.8732±0.1182 | 0.8909±0.0927 | 0.8203±0.1212 | 0.8987±0.0868 | 0.8453±0.1085 | 0.8684±0.1167 |
> | RAP(LLM+pUCT)     | 0.1391±0.0318  | 0.2967±0.0347  | 0.3185±0.01   | 0.3205±0.0174 | 0.3183±0.0046 | 0.3278±0.0101 | 0.3242±0.0122 |
>
> 3. In line with the reviewer's suggestion, we had already compared our work with RAP (LLM+UCT), a method designed for LLM-based planning. Our evaluation shows that RAP's performance drops significantly in stochastic environments (Figure 1). Critically, and as detailed in Figures 3, 4, and Table 1, our method significantly outperforms RAP. This performance advantage also holds against the RAP variants incorporating uncertainty-dealing mechanisms, RAP-D and RAP-E (Figures 3, 4, Table 1).
>
> > W1: Mischaracterizing a design choice as a fundamental limitation: ...LLM-based MCTS baselines...not designed for stochastic outcomes...It is not...a fundamental limitation of the MCTS framework...
>
> 4. **It is a limitation when applying vanilla MCTS for LLM even if the environment is deterministic**. As we have stated in the introduction, LLM planning suffers from LLM generation uncertainty, when using LLM as the world model (as we do in this work), LLM may generate different text descriptions even for the same state. This may be interpreted as different stochastic states. Moreover, state aliasing (e.g., caused by partial observability) could effectively introduce environmental stochasticity for a deterministic environment.
>
> 5. **We agree with the reviewer that the drawback of vanilla MCTS can be viewed as a "simplifying implementation choice"**. For example, Sparse UCT [2] can be viewed as a non-simplified version of UCT for stochastic environments. We will add such discussion into the main text and present the results of MCTS with modeling stochastic environment outcomes (Sparse UCT).
>
> [1]Improved monte-carlo search. Univ. Tartu, Estonia, Tech. Rep, 2006
>
> [2]Lower bounding Klondike solitaire with Monte-Carlo planning, Conf. Autom. Plan. 2009
>
> [3]Combining online and offline knowledge in UCT. ICML 2007
>
> >W2: Quality of evaluation: ...Introducing stochasticity by...having a random chance of an action failing...on problems with genuine, inherent uncertainty.
>
> >L1: Generalization and scope: The evaluation is confined to planning tasks with artificially injected, simplistic stochasticity... more complex and naturalistic uncertainty.
>
> In this rebuttal, we evaluate PlanU on two real-world planning benchmarks: a travel planning benchmark and a web agent benchmark using more realistic failure models. Please refer them to our response to Reviewer c3qH.
>
> >W3: The rationale for initializing the value distribution Z during the expansion phase using the policy's sampling probability is not justified...
>
> >Q2: ...specific choice of initializing the value distribution Z based on the policy's action probability during expansion? An ablation study...
>
> The proposed initialization serves as a **warm start** for the policy. We have studied different initialization strategies: random and uniform. The random choice initialize the distribution randomly, whereas the uniform choice assign a uniformly spaced set of points within the interval [0, 1] to the distribution. Their results, obtained from the Blockworld experiment where Mistral-7B was used as the underlying LLM, are listed as follows. It is shown that using the prior from LLM for initilization can significantly speed up learning.
>
> **Blockworld**
> |Method|2-step|4-step|6-step|
> |-|-|-|-|
> |RAP|0.892±0.03|0.514±0.06|0.166±0.04|
> |PlanU_random|0.972±0.02|0.632±0.03|0.324±0.03|
> |PlanU_uniform|0.946±0.03|0.553±0.05|0.359±0.03|
> |PlanU|1.000±0.00|0.803±0.04|0.559±0.04|
>
>
> For closed-source model the prior probability of LLM cannot be obtained, we choose initialize the distribution using the uniform strategy. As it is depicted in our response to Reviewer c3qH regarding GPT-4.1 and Gemini-2.5-Pro, PlanU performs better than RAP significantly.
>
> >Q1: ...canonical MCTS algorithms like UCT can handle stochastic environments...why they are insufficient for the problems...directly extend an UCT algorithm to LLMs and use it as a baseline?
>
> We had compared our work with RAP, an algorithm that extend UCT algorithm to LLMs. It performs poorly in stochastic environments on 7-8B LLMs, as depicted in Figure 3, 4 and Table 1. We shows the results between PlanU and RAP on Blocksworld on large LLMs as follows.
>
> **Openai/GPT-4.1**
> |Method|2-step|4-step|6-step|8-step|
> |-|-|-|-|-|
> |RAP|1.000±0.00|0.763±0.05|0.341±0.03|0.231±0.06|
> |PlanU|1.000±0.00|0.821±0.04|0.648±0.03|0.392±0.04|
>
>
> **Google/Gemini-2.5-pro**
> |Method|2-step|4-step|6-step|8-step|
> |-|-|-|-|-|
> |RAP|1.000±0.00|0.776±0.03|0.328±0.04|0.238±0.05|
> |PlanU|1.000±0.00|0.808±0.05|0.662±0.03|0.378±0.04|
>
>
> **Qwen/Qwen2.5-14B-Instruct**
> |Method|2-step|4-step|6-step|8-step|
> |-|-|-|-|-|
> |RAP|0.973±0.02|0.605±0.03|0.219±0.03|0.161±0.05|
> |PlanU|1.000±0.00|0.750±0.02|0.521±0.05|0.182±0.04|
>
>
> >L2:Failure modes: A discussion of potential failure modes would be valuable.
>
> 1. In the WebShop benchmark, we inject internet latency into web-related actions. An action is considered failed if the latency exceeds a predefined timeout period.
>
> 2. For the TravelPlanner benchmark, we simulate real-world uncertainty — a critical challenge in travel planning where transportation delays often disrupt itineraries — by injecting realistic delay probabilities into planes and trains. These probabilities are derived from two sources: empirical probability density function (PDF) modeling [4] and a National Transportation statistics. Specifically, the delay probabilities for flights are as follows:
>
> | Delay Duration Range       | Probability (%) |
> | :------------------------- | :-------------- |
> | On-time (-5 to 5 minutes)  | 80.76|
> | Delayed (> 5 - 15 minutes) | 5.62|
> | 15 - 30 minutes delayed    | 5.54|
> | 30 - 60 minutes delayed    | 5.01|
> | Over 60 minutes delayed    | 3.07|
>
> For trains, the delay probability is set to 2% based on the average of the past 5 years from a National Transportation. Such delays can invalidate subsequent segments of generated travel plans, regarding as failure modes, mirroring how real-world disruptions (e.g., a delayed flight causing missed train connections) impact trip feasibility.
>
> [4]Statistical characterization of airplane delays, Scientific Reports 2021

---

> > ### Comment · Reviewer_bLkk · 2025-08-08
> >
> > Thank you for the detailed rebuttal and for running the extensive new experiments to address my comments.
> >
> > To clarify, my main point was that a more direct validation of your method would be a comparison against MCTS variants designed for stochastic environments (see [1, 2] for example). I strongly encourage you to include the new results comparing against Sparse UCT in the revised paper. The new experiments on the travel and web agent benchmarks are also excellent additions that should be included.
> >
> > Your response has addressed my main concerns, and I have adjusted my rating accordingly. I do not have any further questions.
> >
> > [1] Trial-based Heuristic Tree Search for Finite Horizon MDPs, Keller and Helmert (2013)
> >
> > [2] Monte Carlo Tree Search for Continuous and Stochastic Sequential Decision Making Problems, Couetoux (2013)

---

> > > ### Author Response · Authors · 2025-08-09
> > >
> > > Dear Reviewer bLkk,
> > >
> > > Thank you for raising your score, for your encouraging feedback, and for your clarification. We will include the results of Sparse UCT, the travel agent, and the web agent benchmark in our work. We will discuss [1][2] in the related work section.
> > >
> > > ​​As we incorporate your suggestions, please let us know if there are any remaining concerns.
> > >
> > > Best Regards,
> > >
> > > Authors of PlanU

---

> > > > ### Author Response · Authors · 2025-08-09
> > > >
> > > > Dear Reviewer bLkk,
> > > >
> > > > Thank you again for increasing the rating. We show more results of Sparse UCT in the following tables to further strengthen the evaluation.
> > > >
> > > > We present the results for Sparse UCT across the Webshop benchmark and the Entertainment environment, alongside the Tomato salad and Tomato lettuce salad environments. These new results confirm PlanU’s consistent advantage over MCTS variants (including Sparse UCT) and other strong baselines across different domains (web agent and 3D/2D embodied AI).
> > > >
> > > > **Webshop**
> > > > |Method|Average reward|Success rate|
> > > > |-|-|-|
> > > > |CoT|0.46±0.04|0.1|
> > > > |RAP|0.41±0.02|0.2|
> > > > |LATS|0.57±0.07|0.3|
> > > > |PlanU|0.73±0.07|0.5|
> > > > |LLM+Sparse UCT|0.46±0.06|0.2|
> > > >
> > > > **Entertainment**
> > > > |                  | 0              | 1000          | 2000          | 3000          | 4000          | 5000          | 6000          |
> > > > | ---------------- | -------------- | ------------- | ------------- | ------------- | ------------- | ------------- | ------------- |
> > > > | PlanU            | 0.0084±0.0044  | 0.46±0.2798   | 0.877±0.0911  | 0.9453±0.073  | 0.9487±0.0575 | 0.9914±0.0851 | 0.9822±0.0114 |
> > > > | RAP              | -0.0041±0.0    | 0.0701±0.0    | 0.094±0.0     | 0.5423±0.2322 | 0.8957±0.0447 | 0.7277±0.0115 | 0.8278±0.0217 |
> > > > | LLM + Sparse UCT | -0.0044±0.0058 | 0.0719±0.0221 | 0.0837±0.0158 | 0.0634±0.0118 | 0.1396±0.0194 | 0.7824±0.1666 | 0.8347±0.1853 |
> > > >
> > > > **Tomato salad**
> > > > ||0|1000|2000|3000|4000|5000|6000|
> > > > |-----------------|-------------|-------------|-------------|-------------|-------------|-------------|-------------|
> > > > |PlanU|0.2941±0.0099|0.3696±0.0335|0.9644±0.2569|1.0568±0.0709|1.1509±0.1067|1.11±0.0978|1.2079±0.0324|
> > > > |RAP (LLM+UCT)|0.0516±0.0128|0.3675±0.0729|0.4756±0.0678|0.2687±0.0736|0.679±0.0618|0.5844±0.064|0.2588±0.0804|
> > > > |LLM+Sparse UCT|0.1053±0.0469|0.2046±0.073|0.5546±0.0836|0.5415±0.0748|0.5491±0.0816|0.5522±0.0809|0.5453±0.076|
> > > > |RAP(LLM+pUCT)|-0.0645±0.0588|-0.0143±0.0878|0.9895±0.0755|0.941±0.0662|0.9678±0.0533|0.9868±0.058|0.9603±0.0453|
> > > >
> > > >
> > > > **Tomato lettuce salad**
> > > > |                   | 0             | 1000          | 2000          | 3000          | 4000          | 5000          | 6000          |
> > > > | ----------------- | ------------- | ------------- | ------------- | ------------- | ------------- | ------------- | ------------- |
> > > > | PlanU             | -0.083±0.0415 | 0.8995±0.0832 | 1.1069±0.0032 | 1.096±0.0132  | 1.1045±0.0032 | 1.0955±0.0173 | 1.105±0.0039  |
> > > > | RAP (LLM+UCT)      | -0.0315±0.0268 | -0.0374±0.0362 | 0.3184±0.0847 | 0.245±0.0826  | 0.2676±0.0812 | 0.3081±0.0786 | 0.2741±0.0786 |
> > > > | LLM+Sparse UCT | 0.0142±0.0538 | 0.8732±0.1182 | 0.8909±0.0927 | 0.8203±0.1212 | 0.8987±0.0868 | 0.8453±0.1085 | 0.8684±0.1167 |
> > > > | RAP(LLM+pUCT)     | 0.1391±0.0318  | 0.2967±0.0347  | 0.3185±0.01   | 0.3205±0.0174 | 0.3183±0.0046 | 0.3278±0.0101 | 0.3242±0.0122 |
> > > >
> > > > We hope these additional experimental results provide a comprehensive validation of our approach.
> > > >
> > > > Best Regards,
> > > >
> > > > Authors of PlanU

---

> ### Author Response · Authors · 2025-08-04
>
> Dear Reviewer bLkk,
>
> Thanks for your effort in reviewing our work. To help us address your concerns most effectively, could you kindly specify any remaining points requiring further clarification?
>
> We have addressed your concerns in the rebuttal. We summarize our response as follows.
>
> 1. We addressed your concerns regarding the **suitability of MCTS for stochastic environments** (W1, Q1, Q2) by clarifying the difference between action stochasticity and environmental stochasticity. We further compared PlanU with canonical UCT, Sparse UCT, and pUCT in both synthetic and real-world environments, demonstrating PlanU's superior performance.
>
> 2. We addressed your concerns regarding the **initialization of the value distribution Z** (Q2, W3) by conducting an ablation study comparing different initialization strategies. Results show that using LLM priors leads to faster convergence and better performance.
>
> 3. We addressed your concerns regarding the **realism of stochasticity** (W2, L1, L2) by introducing two real-world planning environments: WebShop and TravelPlanner. These benchmarks include realistic failure modes, such as latency-based action failures and probabilistic transportation delays derived from empirical data and national statistics.
>
> In summary, based on your feedback we added:
>
> 1. Ablation studies on value distribution initialization to validate the design choice。
> 2. Two new benchmarks (WebShop, TravelPlanner) with empirically grounded stochasticity and realistic failure modes.
> 3. Extended experiments using GPT-4.1, Gemini-2.5-Pro, and Qwen2.5-14B, showing that PlanU consistently outperforms RAP even in high-performance LLM settings.
>
> During the rebuttal, we have also added:
>
> 1. A discussion of Sparse UCT and its conceptual relation to PlanU, and additional experimental results to support this comparison.
> 2. Compute cost and token/runtime experiments
> 3. Experiments on different operater and different setence-transformer models.
> 4. Abalations experiments of prompt perturbation between PlanU and RAP.
>
> **Do you have any remaining concerns? We are eager to discuss them with you.**
>
> Best Regards,
> Authors

---

> > ### Author Response · Authors · 2025-08-06
> >
> > Dear Reviewer bLkk,
> >
> > Thank you again for your thoughtful review. We have summarized and addressed all your concerns in the rebuttal, including additional experiments and benchmarks.
> >
> > Could you kindly let us know if there are any remaining points requiring clarification? We would be happy to provide further details if needed.
> >
> > Best regards,
> >
> > Authors

---

> > > ### Author Response · Authors · 2025-08-08
> > >
> > > Dear Reviewer bLkk,
> > >
> > > There is only 1 day remaining until the new deadline for author-reviewer discussions. We kindly ask you if there are any remaining concerns. If we have addressed all your concerns, can you please reconsider your rating of this work?
> > >
> > > Sincerely,
> > >
> > > Authors of PlanU

---

### Official Review · Reviewer_c3qH · 2025-07-22

**Clarity:** 2
**Significance:** 2
**Originality:** 2
**Rating:** 4
**Confidence:** 3

**Summary:**

The paper introduces PlanU, an MCTS-based decision-making framework for LLM agents that explicitly handles both LLM generation uncertainty and environmental stochasticity. It models each action-node return as a quantile distribution instead of a mean value, and guides exploration with an Upper Confidence Bounds with Curiosity (UCC) score that combines distributional value estimates and a novelty bonus computed via random-network distillation on text embeddings. PlanU is evaluated on stochastic variants of Blocksworld, Overcooked, and VirtualHome.

**Questions:**

1. Can you report wall-clock time, tokens for PlanU vs RAP/ToT?
2. Can PlanU perform well on some real-world tasks?
3. Can PlanU been adapted to Web agent?
4. I think it'll be better if the authors include PlanU's performance into figure 1.
5. Can PlanU been extended to closed-source models?
6. Can you provide a deeper comparison between PlanU and RAP since they are all MCTS-based methods?

**Ethical Concerns:**

["NO or VERY MINOR ethics concerns only"]

**Final Justification:**

My concerns about more experiments have been mostly addressed.

**Limitations:**

yes

**Quality:**

2

**Strengths And Weaknesses:**

Strength:
1. The motivation is clear.
2. The clarity is good and the paper provides intuitive figures to help understanding.

Weakness:
1. The three datasets the paper use are toy, textual and simulated. There're no real-world, long-horizon tasks with rich observations or actuation. And the model sizes are small.
2. Compute cost and token/runtime overhead haven't been discussed.
3. The paper lacks enough details about experiment setup in the main text.

---

> ### Author Rebuttal · Authors · 2025-07-31
>
> > W1: The three datasets the paper use are toy, textual and simulated. There're no real-world, long-horizon tasks with rich observations or actuation. And the model sizes are small.
>
> > Q2: Can PlanU perform well on some real-world tasks?
> > Q3: Can PlanU been adapted to Web agent?
>
> In this rebuttal, we evaluate PlanU on two real-world planning benchmarks: a travel planning benchmark and a web agent benchmark using Qwen3-14B as the underlying LLM.
>
> 1. Travel Planning Benchmark (TravelPlanner [1]): This benchmark requires an agent to generate a comprehensive plan encompassing transportation, meals, and accommodation based on a user query. The agent can perform various search actions to gather necessary information before making decisions. To simulate real-world uncertainty, we inject delay probabilities for planes and trains. The injecting delay probabilities are derived from the empirical probability density function (PDF) modeling [2] and a National Transportation statistics. Consequently, delays can invalidate subsequent segments of the generated plan. We evaluated 45 tasks of easy/medium/hard difficulty with TravelPlanner. The evaluation metrics includes Task Completion Rate (covers core travel elements) and Constraint Satisfaction Rate (average of satisfying TravelPlanner’s commonsense + hard constraints, e.g., no repeated attractions, meeting budget). Results show PlanU outperforms others (table below).
>
> |Method|Task Completion Rate|Constraint Satisfaction Rate|
> |-|-|-|
> |CoT|0.156±0.030|0.022±0.010|
> |RAP|0.222±0.025|0.044±0.012|
> |PlanU|0.378±0.020|0.222±0.015|
>
> 2. Web Agent Benchmark (WebShop [3]): This benchmark evaluates agents tasked with purchasing products in an online shop with over one million items, meeting specific user requirements (e.g., brand, price, size). Agents interact via web clicks and searches, making decisions based on observed pages and interaction history. Crucially, to mimic real-world conditions, web actions incur network latency following a long-tail log-normal distribution [4] (mean 2s), with actions failing if latency exceeds 10 seconds. Performance is measured by reward (percentage of user-specified attributes satisfied) and success rate (frequency of fully meeting all requirements) across 10 shopping instructions. Experimental results (below) demonstrate that PlanU achieves superior performance compared to several baselines.
>
> |Method|Average reward|Success rate|
> |-|-|-|
> |CoT|0.46±0.04|0.1|
> |RAP|0.41±0.02|0.2|
> |LATS[5]|0.57±0.07|0.3|
> |PlanU|0.73±0.07|0.5|
>
>
> These evaluations on TravelPlanner and WebShop, incorporating realistic uncertainty like transportation delays and internet latency, highlight PlanU's effectiveness in complex, real-world planning scenarios.
>
> [1]TravelPlanner: A Benchmark for Real-World Planning with Language Agents, ICML 2024
>
> [2]Statistical characterization of airplane delays, Scientific Reports 2021
>
> [3]WebShop: Towards Scalable Real-World Web Interaction with Grounded Language Agents, NeurIPS 2022
>
> [4]On the log-normal distribution of network traffic,Physica D: Nonlinear Phenomena
>
> [5]Language agent tree search unifies reasoning, acting, and planning in language models, ICML 2024
>
> > Q5: Can PlanU been extended to closed-source models?
>
> We evaluate in Blocksworld on OpenAI GPT-4.1 and Google Gemini-2.5-Pro. For closed-source models (e.g., GPT-4.1 and Gemini-2.5-Pro), as the token probability cannot be obtained, we initialized the quantiles using a uniform distribution instead of using the token probability. The results are summarized in the following table. The results show that PlanU perform well on closed-source models (GPT-4.1 and Gemini-2.5-Pro). Besides PlanU perform well on a 14B open source models (Qwen2.5-14B).
>
> **Openai/GPT-4.1**
> |Method|2-step|4-step|6-step|8-step|
> |-|-|-|-|-|
> |CoT|0.541±0.07|0.382±0.06|0.221±0.05|0.063±0.04|
> |ToT|0.595±0.08|0.421±0.07|0.241±0.06|0.077±0.05|
> |RAP|1.000±0.00|0.763±0.05|0.341±0.03|0.231±0.06|
> |PlanU|1.000±0.00|0.821±0.04|0.648±0.03|0.392±0.04|
>
> **Google/Gemini-2.5-pro**
> |Method|2-step|4-step|6-step|8-step|
> |-|-|-|-|-|
> |CoT|0.568±0.08|0.395±0.07|0.228±0.06|0.056±0.05|
> |ToT|0.622±0.07|0.408±0.08|0.248±0.07|0.070±0.06|
> |RAP|1.000±0.00|0.776±0.03|0.328±0.04|0.238±0.05|
> |PlanU|1.000±0.00|0.808±0.05|0.662±0.03|0.378±0.04|
>
> **Qwen/Qwen2.5-14B-Instruct**
> |Method|2-step|4-step|6-step|8-step|
> |-|-|-|-|-|
> |CoT|0.405±0.06|0.178±0.05|0.138±0.04|0.028±0.03|
> |ToT|0.486±0.05|0.247±0.06|0.131±0.05|0.042±0.04|
> |RAP|0.973±0.02|0.605±0.03|0.219±0.03|0.161±0.05|
> |PlanU|1.000±0.00|0.750±0.02|0.521±0.05|0.182±0.04|
>
>
> > W2: Compute cost and token/runtime overhead haven't been discussed.
> > Q1: Can you report wall-clock time, tokens for PlanU vs RAP/ToT?
>
> We had placed a detailed analysis of the computational overhead in Appendix C.5. We study the token usages, query times, wall-clock time, and rewards for the Tomato Lettuce Salad task. The results are shown as follows.
>
> |Method|PlanU|PlanU w/o dist|PlanU w/o UCC|RAP|COT|TOT|
> |-|-|-|-|-|-|-|
> |Token Usage(k)|922.69±6.6|1197.99±7.6|1521.86±2.9|2643.84±8.9|35.98±3.2|159.85±4.6|
> |Query times|1389.6±10.0|2353.0±14.9|2289.2±4.5|4313.0±14.8|150.0±0.0|150.0±0.0|
> |Wall clock time(s)|524.47±13.65|989.45±19.81|931.21±17.41|1447.20±25.11|183.46±3.11|206.38±4.95|
> |Rewards|1.1±0.01|0.35±0.02|0.71±0.08|0.26+0.01|0.11+0.00|0.23±0.00|
>
>
> As it is shown, PlanU is highly resource-efficient compared to other MCTS-based methods. It consumes **nearly three times fewer** tokens than RAP, demonstrating its superior ability to navigate the search space effectively. In general, MCTS-based methods performs better than CoT and ToT at the cost of most token consumption. The higher token requirement when compared to non-search methods like CoT and ToT is an expected trade-off, which allows PlanU to robustly handle complex planning problems under uncertainty, a domain where simpler methods like CoT and ToT frequently fail.
>
> >W3: The paper lacks enough details about experiment setup in the main text.
>
> The details of our experimental setup had been **described in Appendix C**. This includes our general setup and computing resources in Appendix C.1. The setup for the Blocksworld, Overcooked, and VirtualHome environment are detailed in Appendix C.2, C.3, and C.4, respectively. Each of these sections covers the complete task description,  observation/action spaces, environment dynamics, and reward structure, ensuring full reproducibility.
>
> >Q4: I think it'll be better if the authors include PlanU's performance into figure 1.
>
> We have put the results of the new Figure 1 as follows. It shows that PlanU performs better than CoT, ToT, and RAP **both in deterministic and stochastic environments**.
>
> **Deterministic**
> |Method|2-step|4-step|6-step|8-step|
> |-|-|-|-|-|
> |CoT|0.541±0.05|0.276±0.04|0.152±0.03|0.028±0.01|
> |ToT|0.541±0.05|0.316±0.05|0.152±0.03|0.063±0.02|
> |RAP|0.973±0.02|0.882±0.03|0.634±0.04|0.357±0.05|
> |PlanU|1.000±0.00|0.921±0.03|0.848±0.03|0.503±0.04|
>
>
> **Stochastic**
> |Method|2-step|4-step|6-step|8-step|
> |--------|---------------|---------------|--------------|--------------|
> |CoT|0.351±0.06|0.237±0.05|0.124±0.04|0.014±0.05|
> |ToT|0.459±0.05|0.263±0.04|0.131±0.03|0.056±0.02|
> |RAP|0.946±0.03|0.553±0.05|0.255±0.04|0.175±0.03|
> |PlanU|1.000±0.00|0.803±0.05|0.524±0.04|0.238±0.03|
>
>
> >Q6: Can you provide a deeper comparison between PlanU and RAP since they are all MCTS-based methods?
>
> We can view PlanU and RAP from four different angles: visual comparison, algorithmic design, performance, and resource consumption.
>
> 1. In Section 5.2, Figure 3 visually describes the difference among the trees used by RAP (written as LLM-MCTS) and PlanU. RAP cannot model the Q value for even such a simple task due to the limitation of vanilla MCTS, whereas PlanU models environmental stochastic using quantile distribution.  We have described the drawbacks of vanilla MCTS in Appendix A2 in detail.
>
> 2. Regarding algorithmic design, PlanU differs from RAP mainly in two aspects:
> (1) **Environmental Uncertainty**— RAP uses the vanilla MCTS with UCT. The vanilla MCT+UCT assumes deterministic state transitions, while PlanU does not assume such simplification. It considers environmental stochastic through  modeling returns as quantile distributions, which better capture environmental uncertainty during planning;
> (2) **LLM uncertainty** — PlanU uses the UCC score, which considers LLM uncertainty, whereas RAP does not explicitly consider such uncertainty.
> 3. Regarding performance, as we have shown in the new Figure 1 in the previous response, *PlanU performs better than RAP both in deterministic and stochastic environments.
> 4. Regarding resource consumption on stochastic environments, **PlanU consumes fewer tokens and wall clock times than RAP**, as we have reported in previous responses. We further report the number of expanded nodes (lower indicates higher efficiency) on Overcooked tasks. PlanU consume a significantly lower cost than RAP, demonstrating both effectiveness and efficiency. Moreover, we show that each component of PlanU contributes its low number of expanded nodes.
>
> | Method|Tomato Lettuce Salad|Tomato Salad|
> |-|-|-|
> | PlanU| 10244±33.7| 5170±10.8 |
> | PlanU w/o Dist| 18920±50.6| 6715±25.4|
> | PlanU w/o UCC| 12783±39.1|5225±14.4|
> | RAP| 26208±64.1| 6945±29.8|

---

> > ### Comment · Reviewer_c3qH · 2025-08-08
> >
> > Thank you for the responses. My concerns about more datasets/extension to closed-form models and token/time overhead have been mostly addressed. So I decide to raise my score to 4.

---

> > > ### Author Response · Authors · 2025-08-08
> > >
> > > Dear Reviewer c3qH,
> > >
> > > Thank you for raising your score and for your encouraging feedback. ​​As we incorporate your suggestions, could you please let us know if there are any remaining concerns?
> > >
> > > Best Regards,
> > >
> > > Authors of PlanU

---

> ### Author Response · Authors · 2025-08-04
>
> Dear Reviewer c3qH,
>
> Thanks for your effort in reviewing our work. We have addressed your concerns in the rebuttal and summarize our response as follows:
>
> 1. Real-world applicability (W1, Q2, Q3): We have conducted evaluations on two real-world tasks: TravelPlanner and WebShop, showing PlanU’s superior performance under realistic uncertainties (delays, latency).
>
> 2. Closed-source adaptation (Q5): We have evaluated PlanU and others on GPT-4.1 and Gemini-2.5-Pro and additionally on a 14B open-source model (Qwen2.5-14B); the results confirm PlanU's strong performance across all settings.
>
> 3. Computational overhead (W2, Q1): We have reported token usage, query counts, and wall-clock time; PlanU is about three times more efficient than RAP.
>
> 4. Experiment details (W3): We have clarified setups in Appendix C for full reproducibility.
>
> 5. Comparison with RAP (Q4, Q6): We have improved Figure 1 accordingly, and find that PlanU consistently outperforms RAP in both deterministic and stochastic settings.
>
> Do you have any remaining concerns? We are eager to discuss them with you.
>
> Best Regards,
>
> Authors

---

> > ### Author Response · Authors · 2025-08-06
> >
> > Dear Reviewer c3qH,
> >
> > Thank you again for your thoughtful review. We have summarized and addressed your concerns in the rebuttal, including additional real-world evaluations, closed-source model results, and computational overhead analysis.
> >
> > Could you kindly let us know if there are any remaining points requiring clarification? We would be glad to provide further details if needed.
> >
> > Best regards,
> >
> > Authors

---

> > > ### Author Response · Authors · 2025-08-08
> > >
> > > Dear Reviewer c3qH,
> > >
> > > There is only 1 day remaining until the new deadline for author-reviewer discussions. We kindly ask you if there are any remaining concerns. If we have addressed all your concerns, can you please reconsider your rating of this work?
> > >
> > > Sincerely,
> > >
> > > Authors of PlanU

---

### Author Response · Authors · 2025-08-07
**Paper Summary**

We sincerely appreciate all reviewers for their insightful feedback, which has significantly strengthened our work. The reviewers acknowledge our work as an impactful extension in LLM-based planning under uncertainty, noting valuable contributions to robust uncertainty handling (bLkk, QuAk, 6Yh1), novel integration of quantile-based value estimation into LLM-driven MCTS (bLkk, 6Yh1), promising research direction (bLkk), clear motivation and intuitive figures (c3qH), and thorough experiments validating effectiveness over baselines (QuAk, 6Yh1). Guided by the reviewers' comments on real-world validation, baseline comparisons, and model robustness, we expanded evaluations to real-world tasks, included ReAct-style baselines, validated across model scales, and added hybrid ablations.

**Key Experimental Results**

According to the suggestions of the reviewers, we have improved our work as follows:

1. **Real-world Task Performance (WebShop[1], TravelPlanner[2])**: We have evaluated PlanU on real-world benchmarks with more realistic failure models — e.g., latency-driven action failures (timeout-triggered) in WebShop, and empirically grounded transportation delays in TravelPlanner (flight delays and train delays). We find that PlanU outperforms baselines (RAP[4], ToT[5], ReAct-style agents) in success rate and robustness, validating its applicability in real-world tasks.

2. **Comparison with Strong Baselines**: We have compared PlanU against powerful agents (LATS[6], DeLLMa[7]) and MCTS variants (RAP with uncertainty extensions). We find that PlanU consistently outperforms these baselines.

3. **Performance in Both Deterministic and Stochastic Environments**: PlanU achieves superior performance across both deterministic and stochastic environments.

4. **Efficacy of PlanU vs. Simple Integrations**: Experiments combining existing uncertainty-handling methods (DMCTS, EMCTS) or improved UCT (LLM-Sparse UCT) with RAP (resulting in RAP-D, RAP-E, LLM Sparse UCT), and replacing UCT with UCC score in RAP-D (RAP-D-UCC) show that simple integrations of existing uncertainty-aware methods do not yield effective performance in LLM planning.

5. **Resource Consumption**: We have analyzed token usage, query times, wall-clock time, and rewards. The results show that PlanU is highly resource-efficient among MCTS-based methods, consuming nearly three times fewer tokens than RAP.

6. **Cross-model Robustness**: The evaluations across open-source (Qwen2.5-14B) and closed-source (GPT-4.1, Gemini-2.5-Pro) models show that PlanU is adaptable to diverse LLM backbones.

7. **Ablation Studies**: Experiments isolating contributions of quantile-based value estimation and UCC exploration (e.g., quantile + UCT vs. mean + UCC) show that both components are critical.

8. **Value Distribution Initialization**: Experiments comparing initialization strategies (random, uniform, and LLM policy prior) show that using LLM policy priors for initializing the value distribution Z accelerates the learning process.

9. **Efficacy of Distributional Learning Over Expectation Operators**: Evaluations of PlanU against variants using mean values (PlanU w/o dist) and different operators ($\tau_{0.5}$, $\mathbb{E}[Z]$-variance, etc) show that learning full value distributions preserves multimodality, leading to more stable performance.

10. **Handling LLM Prompt Uncertainty**: Experiments on prompt shuffling (randomly reordered task descriptions) and injection (with irrelevant text) demonstrate PlanU’s robustness to LLM generation uncertainty.

11. **Sensitivity to Text Encoders and Quantile Number**: The default PlanU outperforms encoder variants. The default n=51 quantiles outperform n=41 and n=61 variants.

12. **Clarity**: We have described the difference between ours and vanilla MCTS in detail. We have compared ours with RAP from design, performance, and resource consumption angles. We have described the experimental setup in Appendix C.


**Core Takeaways**

- PlanU addresses the uncertainty problem in LLM planning through explicitly modeling value distributions (via quantiles) in MCTS and combining them with curiosity-driven exploration (UCC). Such an approach achieves superior performance for uncertain environments across real-world tasks, diverse LLM backbones, and high-stochasticity scenarios.


**References**

[1]TravelPlanner: A Benchmark for Real-World Planning with Language Agents, ICML 2024.

[2]WebShop: Towards Scalable Real-World Web Interaction with Grounded Language Agents, NeurIPS 2022.

[3]Statistical characterization of airplane delays, Scientific Reports 2021.

[4]Reasoning with language model is planning with world model, EMNLP23.

[5]Tree of Thoughts: Deliberate Problem Solving with Large Language Models, NeurIPS 2023.

[6]Language agent tree search unifies reasoning, acting, and planning in language models, ICML 2024.

[7]DeLLMa: A framework for decision making under uncertainty with large language models, ICLR 2025.

---

### Note · Authors · 2025-08-12

We sincerely thank all reviewers for their insightful feedback, which has substantially strengthened this work. Our study introduces **PlanU**, an LLM-based planning method under both **environmental stochasticity** and **LLM generation uncertainty**, enhancing Monte Carlo Tree Search (MCTS) via (i) **quantile-based value distribution modeling** to preserve multimodality in returns, and (ii) a **UCC score** to guide curiosity-driven exploration under uncertainty.

**Key discussion outcomes:**
- Reviewers recognized contributions to **robust uncertainty handling** (bLkk, QuAk, 6Yh1), **novel integration of distributional value estimation into LLM-driven MCTS** (bLkk, 6Yh1), promising research direction (bLkk), clear motivation and figures (c3qH), and thorough experiments (QuAk, 6Yh1).
- Two reviewers (c3qH, bLkk) actively engaged during the discussion and **raised their ratings** after our clarifications and additional experiments. We are grateful for their constructive dialogue.
- We regret that we were unable to discuss with Reviewers QuAk and 6Yh1 during the discussion phase; we hope our extensive updates also address their concerns.

**Major improvements during rebuttal:**
1. **Real-world evaluations** (WebShop, TravelPlanner) with realistic failure models (latency, transportation delays) — PlanU outperforms RAP, CoT, and LATS in WebShop, outperforms RAP, CoT in TravelPlanner.
2. **Strong baseline comparisons** against LATS, DeLLMa, and stochastic MCTS variants — PlanU consistently leads.
3. **Broad environment coverage** — deterministic & stochastic domains, with superior performance in both.
4. **Ablations** showing the necessity of both quantile estimation and UCC exploration.
5. **Cross-model robustness** — effective with Qwen2.5-14B, GPT-4.1, and Gemini-2.5-Pro.
6. **Resource efficiency** — 3× fewer tokens than RAP, competitive wall-clock time.
7. **Robustness to LLM prompt perturbations** — stable under shuffled or injected prompts.
8. **Sensitivity analysis** — optimal at n=51 quantiles; LLM policy priors accelerate learning.

**Takeaway:** PlanU offers a principled approach to LLM planning under uncertainty, validated across **diverse tasks, strong baselines, and model scales**, with significant efficiency gains. We believe these results address all major concerns raised.

We sincerely thank the AC and SAC for their hard work in organizing the review process.

---

### Decision · Program_Chairs · 2025-09-17

**Decision:**

Accept (poster)

**Comment:**

The paper considers decision making under uncertainty, such as planning in stochastic environment; proposes PlanU, an LLM-based planning method that captures uncertainty within MCTS. PlanU models the return of each node in the MCTS as a quantile distribution, which uses a set of quantiles to represent the return distribution. To balance exploration and exploitation during tree search, PlanU introduces an UCC score which estimates the uncertainty of MCTS nodes.

Strength

The paper addresses a timely and important problem. considers various scenarios both in deterministic and stochastic settings. considers a number of benchmarks. includes ablation studies.

The compute cost and experiment details were added during discussion period. The authors also added comparison to other MCTS forms.

The authors added various experiments and addressed the *meaningful concerns*.
I removed reviewer QuAK due to lack of engagement in the discussion process (despite my various efforts) from the final judgement call.

I ask the authors to please add the experiments, comparisons and clarifications that were discussed during the discussion period to the final version of the paper.